# Pancreatic islet chromatin accessibility and conformation reveals distal enhancer networks of type 2 diabetes risk

William W. Greenwald[1,11], Joshua Chiou [2,11], Jian Yan[3,4,11], Yunjiang Qiu [1,3,11], Ning Dai[5,11], Allen Wang[6,7], Naoki Nariai[6], Anthony Aylward[1], Jee Yun Han[7], Nikita Kadakia [6], Laura Regue[5], Mei-Lin Okino[6], Frauke Drees[6], Dana Kramer[8], Nicholas Vinckier[6], Liliana Minichiello [8,9], David Gorkin [7], Joseph Avruch[5], Kelly A. Frazer[6,12], Maike Sander[6,10,12], Bing Ren [3,7,10,12] & Kyle J. Gaulton [6,12]

Genetic variants affecting pancreatic islet enhancers are central to T2D risk, but the gene targets of islet enhancer activity are largely unknown. We generate a high-resolution map of islet chromatin loops using Hi-C assays in three islet samples and use loops to annotate target genes of islet enhancers defined using ATAC-seq and published ChIP-seq data. We identify candidate target genes for thousands of islet enhancers, and find that enhancer looping is correlated with islet-specific gene expression. We fine-map T2D risk variants affecting islet enhancers, and find that candidate target genes of these variants defined using chromatin looping and eQTL mapping are enriched in protein transport and secretion pathways. At *IGF2BP2*, a fine-mapped T2D variant reduces islet enhancer activity and *IGF2BP2* expression, and conditional inactivation of *IGF2BP2* in mouse islets impairs glucose-stimulated insulin secretion. Our findings provide a resource for studying islet enhancer function and identifying genes involved in T2D risk.

[1] Bioinformatics and Systems Biology Graduate Program, UC San Diego, 9500 Gilman Drive, La Jolla, CA 92093, USA. [2] Biomedical Sciences Graduate Program, UC San Diego, 9500 Gilman Drive, La Jolla, CA 92093, USA. [3] Ludwig Institute for Cancer Research, 9500 Gilman Drive, La Jolla, CA 92093, USA. [4] Department of Medical Biochemistry and Biophysics, Division of Functional Genomics and Systems Biology, Karolinska Institutet, SE-171 77 Stockholm, Sweden. [5] Department of Molecular and Cellular Biology, Harvard University, 52 Oxford Street, Cambridge, MA 02138, USA. [6] Department of Pediatrics, UC San Diego, 9500 Gilman Drive, La Jolla, CA 92093, USA. [7] Center for Epigenomics, UC San Diego, 9500 Gilman Drive, La Jolla, CA 92093, USA. [8] European Molecular Biology Laboratory, Mouse Biology Unit, Via Ramarini 32, 00015 Monterotondo, Italy. [9] Department of Pharmacology, University of Oxford, OX1 3QT Oxford, UK. [10] Department of Cellular and Molecular Medicine, UC San Diego, 9500 Gilman Drive, La Jolla, CA 92093, USA. [11] These authors contributed equally: William W. Greenwald, Joshua Chiou, Jian Yan, Yunjiang Qiu, Ning Dai. [12] These authors jointly supervised this work: Kelly A. Frazer, Maike Sander, Bing Ren, Kyle J. Gaulton. Correspondence and requests for materials should be addressed to K.J.G. (email: kgaulton@ucsd.edu)

Genetic risk of type 2 diabetes (T2D) is largely mediated through variants affecting enhancer activity in pancreatic islets[1–7]. The genes regulated by islet enhancers are largely unknown, however, impeding discovery of disease-relevant gene networks and pathways perturbed by risk variants and the development of novel therapeutic avenues. The spatial organization of chromatin plays a critical role in tissue-specific gene regulation, and recently developed high-throughput techniques, such as Hi-C enable the characterization of physical relationships between genomic regions in human tissues genome-wide[8–11]. Tissue-specific maps of chromatin conformation can be used to identify candidate target genes of distal regulatory elements and inform the molecular mechanisms of disease risk variants[9]. A map of chromatin conformation in islets could thus facilitate the annotation of islet regulatory networks and help elucidate the molecular mechanisms of T2D risk loci and the gene networks they affect.

In this study, we generate a high-resolution, genome-wide map of three-dimensional (3D) chromatin architecture in pancreatic islets, and use this map to annotate candidate target genes of islet enhancers defined using ATAC-seq assays and published ChIP-seq data. We identify distal candidate target genes for thousands of islet enhancers, many of which interact over 1 MB, and find that genes with enhancer interactions correlate with islet-specific expression. We identify 30 T2D risk signals with fine-mapped variants in islet enhancers, of which 24 have significant allelic imbalance in islet accessible chromatin. Candidate target genes of these T2D enhancer signals, defined by combining chromatin looping and promoter-proximity with eQTL mapping, are specifically enriched in protein secretion and transport pathways. Finally, at the *IGF2BP2* locus, we show that T2D risk alleles reduce islet chromatin accessibility and expression of target gene *IGF2BP2* and that conditional knockout of *IGF2BP2* homolog *Imp2* in mouse islets impairs glucose-stimulated insulin secretion. Altogether our results provide target genes of islet enhancer activity, through which we link islet enhancer regulation of protein transport and secretion pathways to genetic risk of T2D.

## Results

**Islet chromatin accessibility and 3D chromatin architecture**. We first defined islet accessible chromatin using ATAC-seq[12] generated from four pancreatic islet samples (Supplementary Table 1). We called sites for each sample separately using MACS2[13], and merged sites to create a combined set of 105,734 islet accessible chromatin sites. We observed strong correlation in both accessible chromatin signal and peak calls across samples (Supplementary Fig. 1a), as well as concordance with peak calls from the majority of published ATAC-seq data from 19 islet samples and FACS-sorted beta and alpha cells[7,14,15] (Supplementary Fig. 1b, c). We collected previously published ChIP-seq data of histone modification and transcription factor binding in primary islets from two studies[4,5] and utilized these data to call chromatin states with ChromHMM[16] (Supplementary Fig. 1d). Accessible chromatin predominantly mapped within active enhancer (EnhA1) and promoter (TssA) states (Fig. 1a). We functionally annotated islet accessible chromatin peaks using chromatin states to define active enhancers and promoters, as well as other classes of islet accessible chromatin (Supplementary Data 1). We identified 44,860 active enhancers which, in line with previous reports[4,17], were distal to promoters (Supplementary Fig. 1e), more tissue-specific (Supplementary Fig. 1f), overlapped islet transcription factor ChIP-seq sites (Supplementary Fig. 1g), and preferentially harbored sequence motifs for FOXA, RFX, NEUROD, and other islet transcription factors (Supplementary

Data 2). These results define active enhancers and other classes of accessible chromatin in pancreatic islets.

Defining the target genes of enhancers has been challenging as they frequently control non-adjacent genes over large genomic distances through chromatin looping[18]. To address this, we created a map of 3D chromatin architecture in pancreatic islets at sufficient resolution to identify chromatin loops. We performed genome-wide chromatin conformation capture using in situ Hi-C[8,19] in three islet samples, two of which were sequenced to a depth of >1 billion reads (Supplementary Table 1). Contact matrices from islet Hi-C assays were strongly correlated across samples (Spearman $\rho > 0.80$) (Supplementary Fig. 2a). We called chromatin loops at 5, 10, and 25 kb resolution with HICCUPS[8] using reads from each sample individually, as well as with reads pooled from all three samples (Fig. 1b). We merged the resulting four sets of loop calls where both anchors overlapped at 20 kb resolution (see Methods) to create a combined set of 11,924 islet Hi-C loops (Supplementary Data 3). The median distance between loop anchor midpoints was 255 kb, and nearly 10% were over 1 Mb in size (Supplementary Fig. 2b). This established a map of chromatin loops in human pancreatic islets.

We next determined the relationship between islet accessible chromatin and chromatin looping. Islet accessible chromatin signal was largely localized to islet loop anchors, with the strongest signal at anchor midpoints (Fig. 1c). Nearly half (48.7%) of all islet accessible chromatin sites were within 25 kb of an anchor, and 16.8% directly overlapped an anchor. Sites most enriched (empirical $P < 1.5 \times 10^{-4}$) for direct overlap with chromatin loop anchors were those in a CTCF-binding state (7.5-fold), followed by active promoter (TssA: 3.9-fold; TssFlnk: 3.3-fold), and active enhancer (EnhA1: 2.4-fold) states (Fig. 1d). We further mapped the relationship between pairs of islet accessible chromatin sites directly connected by loop anchors (Supplementary Fig. 2c). The most significantly enriched anchor interactions were between active enhancer and promoter states (EnhA1-TssA OR = 1.28, Fisher's exact $P = 1.53 \times 10^{-37}$; EnhA1-EnhA1 OR = 1.37, $P = 1.87 \times 10^{-38}$; TssA-TssA OR = 1.42, $P = 6.15 \times 10^{-36}$). We also observed strong enrichment for interactions between sites within the CTCF-binding state (CTCF-CTCF OR = 1.16; Fisher's exact $P = 1.1 \times 10^{-17}$) (Fig. 1e). These results demonstrate that islet chromatin loops are prominently enriched for CTCF binding, as well as active promoter and enhancer regions.

**Enhancer loops and islet-specific gene expression**. We next used chromatin loops to annotate candidate relationships between distal islet enhancers and their potential target genes genome-wide (see Methods). We identified 6278 islet active enhancers that mapped directly in a chromatin loop anchor and, of these, 3022 enhancers were in a loop to a gene promoter (Supplementary Fig. 2d and Supplementary Data 4). Conversely, the promoter regions of 2028 genes had at least one direct loop to an active enhancer element (Supplementary Fig. 2e and Supplementary Data 5). Of these 2028 genes, 952 (47%) had chromatin loops to multiple active enhancers (Supplementary Fig. 2e). Genes directly looped to multiple enhancers were enriched for processes related to transcription factor activity and gene regulation, signaling and stimulus response, protein transport and insulin signaling (Supplementary Table 2), and also included genes critical for islet function such as *ISL1, FOXA2, NKX6.1,* and *MAFB* (Supplementary Data 5). At many loci enhancers looped to gene promoters over long distances; the average distance between interacting enhancer and gene promoter pairs was 165 kb, with 13.9% (532) over 500 kb and 3.6% (138) over 1 Mb (Fig. 2a). For example, there were four chromatin loops at the *MAFB* locus,

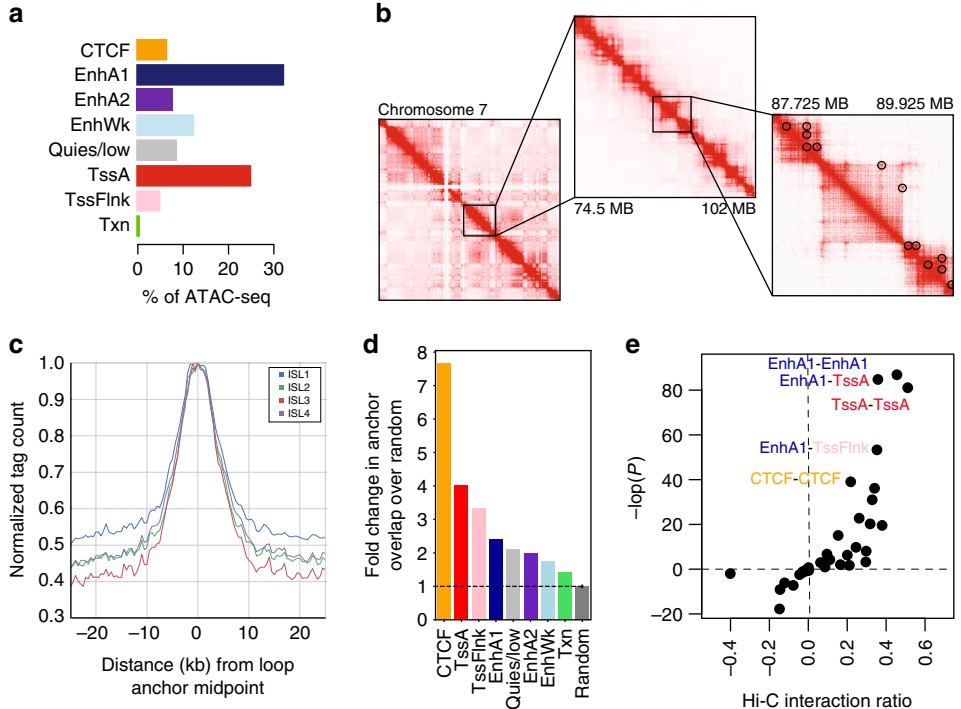

**Fig. 1** Chromatin accessibility and conformation in pancreatic islets. **a** Islet accessible chromatin signal mapped predominantly within active enhancer (EnhA1) and promoter (TssA) states. **b** Chromatin looping from in situ Hi-C assays of three pancreatic islet samples at entire chromosome (left), 25 MB (middle) and 2 MB (right) resolution on chromosome 7. Black circles on the right panel represent statistically significant loop calls. **c** Accessible chromatin signal from four islet samples (ISL1-4) was distributed around chromatin loop anchor midpoints. **d** Islet chromatin loop anchors were enriched for islet CTCF-binding sites, as well as active enhancers and active promoters compared to random sites. Values represent fold change, and the error bar is SD. **e** Islet chromatin loops were most enriched for interactions between islet active enhancers and active promoter elements, and between CTCF-binding sites

including two direct loops between enhancers and the *MAFB* promoter region over 1 Mb distal (Fig. 2b). These results define candidate target genes for thousands of distal enhancer elements in islets.

We examined the relationship between active enhancer looping and target gene expression. We compared our map of islet enhancer candidate target genes defined from islet chromatin loops to gene expression levels in independent RNA-seq data from pancreatic islet samples[20] and 53 tissues in GTEx release v7 data[21]. A significantly higher proportion of genes expressed in islets had at least one enhancer loop compared to non-islet expressed genes (ln (TPM) >1; expr = 0.13, non-expr = 0.05, $\chi^2$ $P < 2.2 \times 10^{-16}$). Genes with increasing numbers of enhancer loops had, on average, higher expression level in islets (Spearman $\rho = 0.13$, $P < 2.2 \times 10^{-16}$), with the highest expression among genes with six or more loops (median = 19.1 TPM) (Fig. 2c). We measured the relative expression level of genes in islets and 53 GTEx tissues normalized across tissues (see Methods), and again observed a significant relationship between enhancer loops and relative islet expression level (Spearman $\rho = 0.084$, $P < 2.2 \times 10^{-16}$) (Fig. 2d). In addition, the number of islet enhancer interactions was a significant predictor of higher relative gene expression level in islets (linear regression $\beta = 0.14$, $P < 2.2 \times 10^{-16}$) but not of relative expression level in the 53 other tissues (Fig. 2d). We observed similar correlations between distal enhancers and islet gene expression when considering sites within a 25 kb region around each loop anchor, suggesting that these relationships extend beyond anchor boundaries (Supplementary Fig. 2f, g). These results suggest that distal islet enhancer chromatin loops are correlated with islet-specific gene expression patterns.

We next determined the effects of genetic variants in islet enhancers on target gene regulation. We generated expression

quantitative trait locus (eQTL) data from 230 islet RNA-seq samples by combining summary statistics from two published studies through meta-analysis[7,20] (see Methods). We identified variants overlapping classes of islet regulatory elements genome-wide. We then quantified the eQTL association of these variants to target genes determined by their proximity to nearby genes and from chromatin loops (see Methods). As expected, we observed the strongest eQTL evidence for active promoter and enhancer variants proximal to genes (TssA: median $-\log10(P) = 0.64$; EnhA proximal: median $-\log10(P) = 0.50$) (Fig. 2e). For variants in distal enhancers, we observed significantly stronger evidence for islet eQTL association with genes in direct loops to the enhancer relative to non-loop genes (EnhA loop median = 0.35, EnhA non-loop median = 0.32, Wilcox $P = 4.4 \times 10^{-5}$), even when matching based on gene distance to the enhancer (EnhA non-loop matched, Wilcox $P = 0.022$) (Fig. 2e). We observed similar eQTL enrichment among enhancer variants looped to gene promoters when considering sites within 25 kb of a loop anchor (Supplementary Fig. 2h). These results suggest that genetic variants in distal islet enhancer elements are preferentially correlated with the expression level of genes in chromatin loops.

**Fine-mapped T2D risk signals affect islet enhancer activity.** Genetic variants in islet regulatory elements are enriched for T2D risk[1,2,4,5]. The effects of variants in regulatory elements on T2D risk in the context of chromatin looping, however, are unknown. We determined the effects of variants in islet regulatory elements and chromatin loops on T2D risk using association data of 6.1 M common (MAF > 0.05) variants from the DIAGRAM consortium with fgwas and LD-score regression[22,23]. We observed strongest enrichment of variants in active regulatory elements, most

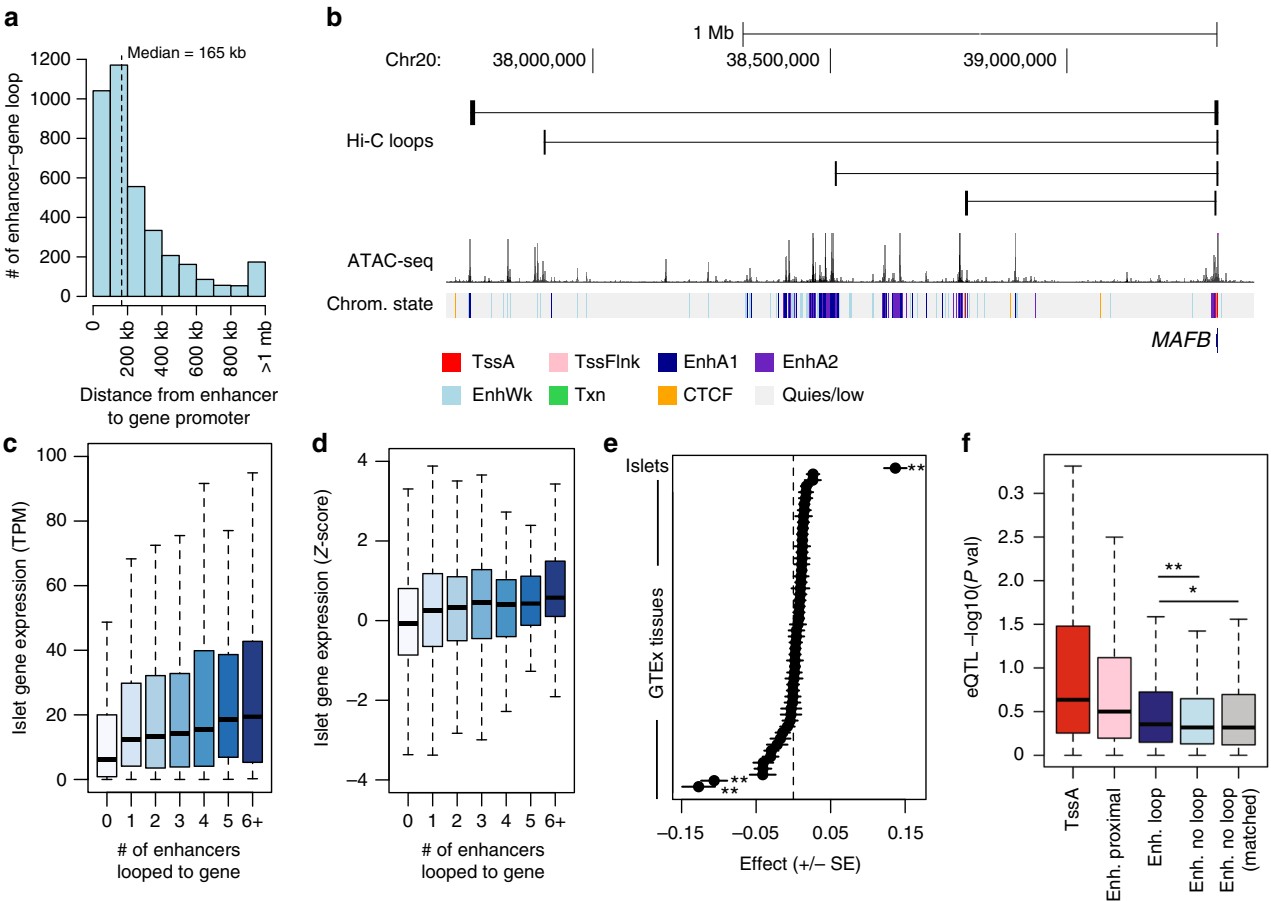

**Fig. 2** Islet enhancer regulation of distal target gene expression. **a** Enhancers looped to gene promoters on average over a 165 kb distance, including >10% over 500 kb. **b** Distal islet enhancers formed chromatin loops with the *MAFB* promoter, including two over 1 MB. **c** Genes with increasing numbers of chromatin loops with islet enhancers had increased expression level in islets, with the highest expression among genes with six or more looped enhancers. **d** Genes with increasing numbers of chromatin loops with islet enhancers in had increased relative expression level in islets Z-score normalized across 53 GTEx tissues. **e** The number of chromatin loops with islet enhancers was a significant predictor of relative islet gene expression compared to relative gene expression in 53 other tissues in GTEx. Values represent effect size and SE from the linear model. **P < 0.0001. **f** Gene expression QTL P-values for genetic variants in gene promoters (TssA; red), enhancers proximal to gene promoters (Enh. proximal; pink), enhancers in chromatin loops to the gene promoter (Enh. loop; dark blue), and enhancers not in chromatin loops to the gene promoter for both all enhancers (Enh. no-loop; light blue) and enhancers distance-matched with looped genes (Enh. no-loop matched; gray). Variants in enhancer elements had stronger evidence for islet expression QTLs with genes in loops than genes with no loop, even when matched based on distance. Wilcox test *P < 0.05, **P < 0.0001. Boxplots show the median, and third and first quartiles. Source data are provided as a source data file

notably in active enhancers (EnhA1 fgwas ln(enrich) = 3.9, LD-score $Z$ = 3.1) (Fig. 3a and Supplementary Fig. 3a). The effects of variants in active enhancer and promoter elements on T2D risk were more pronounced among those in chromatin loops (EnhA1 fgwas ln(enrich) = 4.38, LD-score $Z$ = 3.1; TssA fgwas ln (enrich) = 3.03, LD-score $Z$ = 0.86) (Fig. 3b and Supplementary Fig. 3a). Conversely, variants in other islet elements such as flanking promoters and weak enhancers were more enriched outside of loops (Fig. 3b and Supplementary Fig. 3a). To determine the inter-dependence of these effects, we jointly modeled variants in islet regulatory elements on T2D risk, while also including variants in GENCODE coding exons and UTRs. In a joint model, we observed enrichment of variants in islet active enhancer elements (EnhA1 ln(enrich) = 4.04), in addition to flanking promoters (TssFlnk ln(enrich) = 3.77) and coding exons (CDS ln(enrich) = 2.34) (Supplementary Fig. 3b). These results demonstrate genome-wide enrichment of variants in islet active regulatory elements within chromatin loops for T2D risk.

To identify T2D risk signals mapping in islet enhancers, we used the effects from the joint enrichment model as priors on the causal evidence (posterior probability of association; PPA) for

variants at both known T2D loci and genome-wide[1,2,23] (Supplementary Data 6, see Methods). Among 107 known risk signals, variants in islet enhancers accounted for almost a third (29%) of the total probability mass (Fig. 3c). We clustered known risk signals based on annotations at candidate causal variants (see Methods) and identified 30 signals where the causal variant was likely in an islet enhancer (Fig. 3d). The 30 T2D islet enhancer signals were associated with IGTT-based insulin secretion phenotypes significantly more than un-annotated signals (Enh. = 42%, un-annot. = 17%, Chi-square $P = 1 \times 10^{-7}$), supporting a role in islet function[24] (see Methods, Fig. 3e). Fine-mapping including functional priors improved causal variant resolution at these 30 signals, which on average had 3.5 candidate variants overlapping an islet enhancer and an ATAC-seq site from >1 sample (Fig. 3f, Supplementary Data 1, and Supplementary Data 7). The majority of these enhancers were highly reproducible (>50% of samples), active in beta and alpha cells, in low-methylated regions (LMRs), and bound by islet TFs (Supplementary Data 7). At six signals we resolved a single causal enhancer variant, for example rs7732130 (PPA = 98%) at the 5q13 locus near *ZBED3/PDE8B* (Fig. 3g). Outside of known

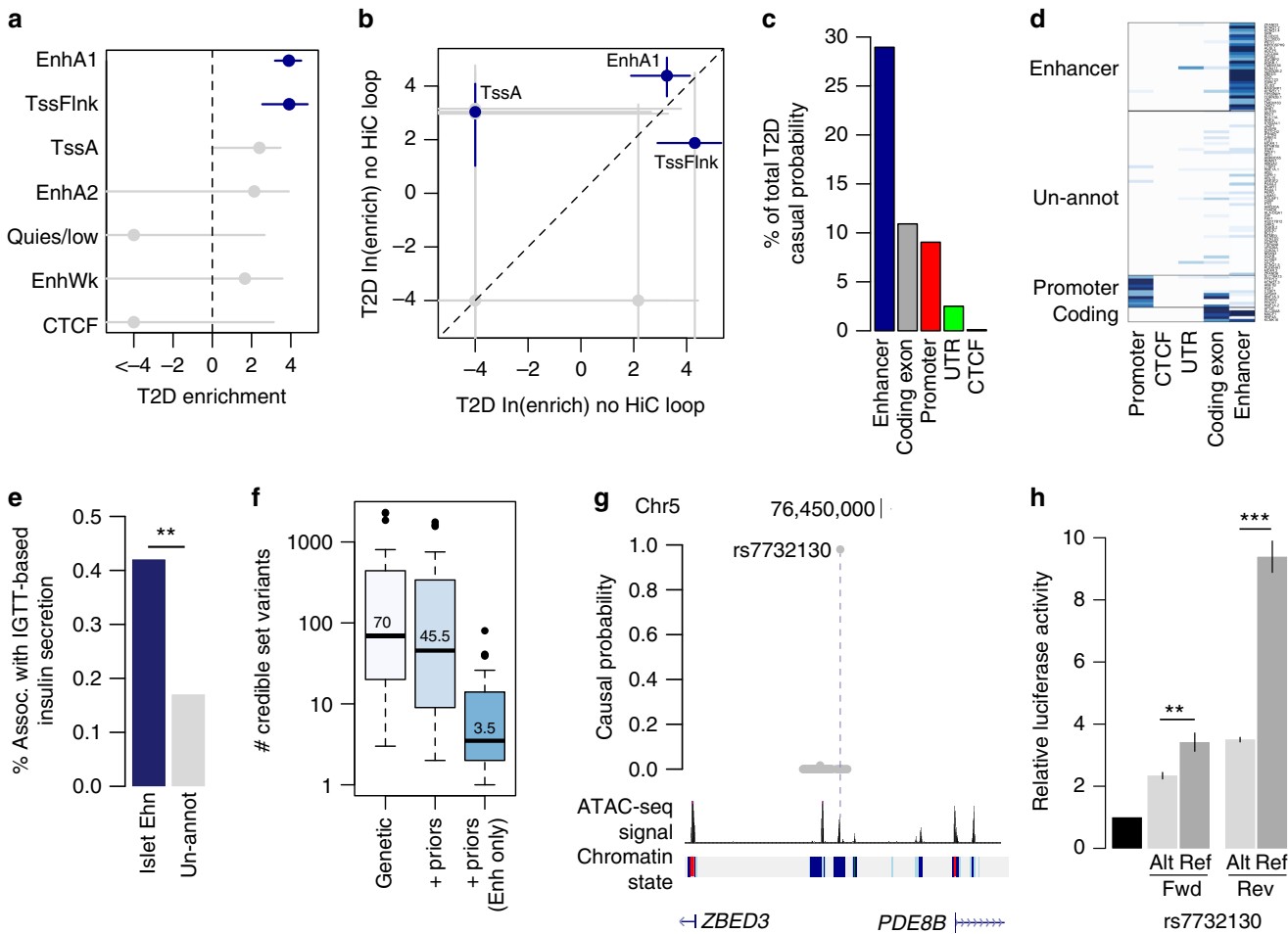

**Fig. 3** Type 2 diabetes risk signals map in islet enhancers. **a** Genetic variants in islet active regulatory elements genome-wide were enriched for T2D risk, with strongest enrichment in active enhancer (EnhA1) elements. Values represent log enrichment and 95% CI, and are colored blue where the 95% CI does not overlap 0. **b** The effects of variants in active enhancer (EnhA1) and promoter (TssA) elements on T2D risk were stronger among those in chromatin loops, whereas other elements were enriched for T2D outside of loops. Values represent log enrichment and 95% CI, and are colored blue where the 95% CI does not overlap 0. **c** Over 30% of the total causal probability across 107 known T2D risk signals mapped in islet enhancer elements. **d** Clustering of known T2D signals based on islet and coding annotations identified 30 signals with likely causal variants in islet enhancers. **e** A significantly higher percentage of T2D islet enhancer signals were associated with IGTT-based insulin secretion phenotypes than un-annotated T2D signals (Chi-square **\*\***$P < 0.001$). **f** Number of variants in the 99% credible sets for the 30 T2D islet enhancer signals based on genetic fine-mapping alone (genetic), genetic fine-mapping, including functional priors (+priors). **g** T2D causal variant rs7732130 at the 5q13 locus near *ZBED3/PDE8B* mapped in an islet active enhancer. **h** rs7732130 has allelic effects on enhancer activity in the islet cell line MIN6 ($N = 3$), where the T2D risk allele and reference (ref) G has higher activity than the alternate (alt) allele A. Values represent mean and SD. *T*-test **\*\***$P < 0.001$, **\*\*\***$P < 0.0001$. Boxplots show the median, and third and first quartiles. Source data are provided as a source data file

loci, we identified an additional 127 loci genome-wide where fine-mapping identified a putative T2D risk variant that overlapped an islet enhancer and ATAC-seq site from >1 sample (Supplementary Fig. 3c, Supplementary Data 1, and Supplementary Data 8; see Methods). These results identify known and putative T2D risk signals with causal variants in islet enhancers.

We next determined allelic effects of variants at these T2D signals on islet enhancer activity. We performed allelic imbalance mapping of enhancer variants using data from 23 islet ATAC-seq samples (four in this study, and 19 from published studies) and three islet Hi-C samples (see Methods). At the 30 T2D enhancer signals, we identified 24 variants with significant allelic imbalance (binomial test; FDR $q < 0.1$) in islet accessible chromatin (median = 1/signal) (Supplementary Data 7). Supporting the function of these variants, we observed significant evidence for concordant direction of effect on allelic imbalance in islet chromatin conformation (binomial $P = 0.022$) and the majority

(19/24) were predicted to disrupt a TF footprint (Supplementary Data 7). Among putative T2D loci, we identified 20 additional variants with significant allelic imbalance (FDR < 0.1) (Supplementary Data 8). T2D variants with significant imbalance included five with previously reported islet regulatory effects such as rs11257655 at 10p13 (binomial $P = 9.1 \times 10^{-7}$), rs11708067 at 3q21 (binomial $P = 2.1 \times 10^{-8}$) and rs10842991 at 12p11 (binomial $P = 2.6 \times 10^{-4}$);[14,25] the former two have also been reported to affect DNA methylation[14,26]. Among the 19 imbalanced variants not reported previously, rs7732130 at 5p13 is causal for T2D (PPA = 98%) and the T2D risk (and reference) allele G increased chromatin accessibility (binomial $P = 7.1 \times 10^{-4}$). We validated that the risk allele at rs7732130 increased islet enhancer activity using gene reporter assays in islet cells (*t*-test Fwd $P = 3.7 \times 10^{-3}$, Rev $P = 6.8 \times 10^{-6}$) (Fig. 3h). These results identify T2D risk variants with allelic effects on islet enhancer activity.

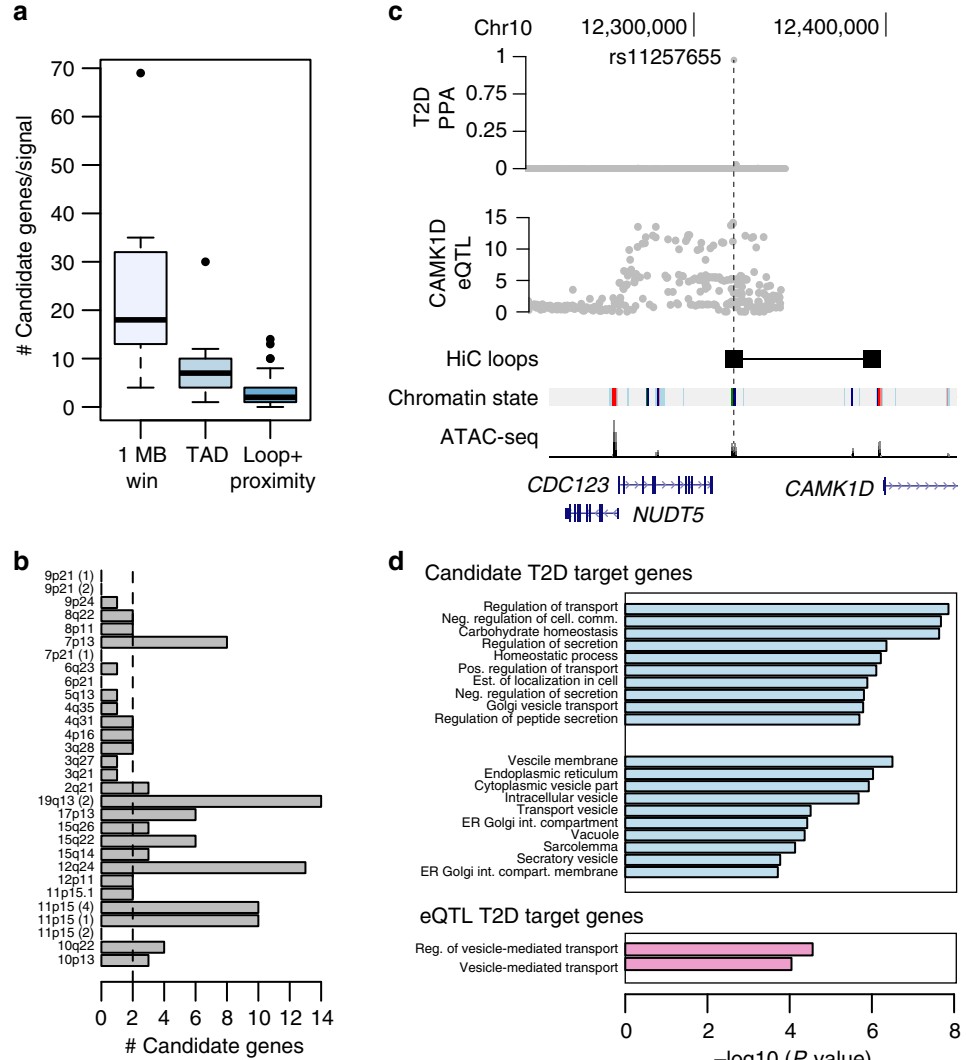

**Fig. 4** Target genes of type 2 diabetes islet enhancer signals are involved in protein secretion and transport. **a** Prioritizing candidate target genes using chromatin loops and promoter-proximity (Loop + proximity; avg = 2) greatly reduces the number of genes obtained when using a 1 MB window (1 Mb win; avg = 18) or topologically associating domain boundaries (TAD; avg = 7). **b** T2D islet enhancer signals formed chromatin loops with, or were in proximity to, an average of two target genes. **c** At the *CDC123/CAMK1D* locus T2D islet enhancer variant rs11257655 was in a chromatin loop to the *CAMK1D* promoter and an islet eQTL for *CAMK1D* expression. Probabilities (PPA) that variants are causal for T2D risk (top) and variant association (-log10 *P*) with islet expression level of *CAMK1D* (middle). **d** Candidate target genes of T2D enhancer signals were strongly enriched for biological processes related to protein secretion, protein transport, vesicles and vesicle membranes, and endoplasmic reticulum (FDR *q* < 0.2) (top), and candidate genes with islet eQTL evidence were specifically enriched for vesicle-mediated transport (FDR *q* < 0.2) (bottom). Boxplots show the median, and third and first quartiles. Source data are provided as a source data file

**Candidate targets of T2D variants affecting islet enhancers.** While a large percentage of T2D risk signals affect islet enhancer activity, the gene targets of these enhancers are unknown. In order to identify genes affected by T2D risk variants in enhancers, we used a tiered strategy whereby we first identified candidate target genes of these enhancers using chromatin looping and promoter-proximity, and then further prioritized candidate genes *cis*-regulated by T2D enhancer variants using eQTL mapping. For each T2D enhancer signal (from Fig. 3d), we identified candidate genes based on whether an enhancer variant was within 25 kb of either a chromatin loop to the gene promoter or the gene promoter itself (see Methods). Based on this definition T2D enhancer signals had on average 2 candidate target genes (Fig. 4a, b and Supplementary Table 3), a large reduction in candidates compared to using a 1 MB window (median = 18 genes) or topologically associating domain (TAD) boundaries (median = 7 genes)

around candidate variants (Fig. 4a). At several loci, loops implicated candidate target genes highly distal (>500 kb) to T2D enhancer variants. For example, at the 3q27 locus T2D variants directly looped to the *TPRG1* promoter 900 kb distal (Supplementary Fig. 4a), and at the 10p13 locus T2D variants looped to the *OPTN* and *CCDC3* promoters 840 kb distal (Supplementary Fig. 4b). In additional examples, T2D enhancer variants at the 11p15 locus near *KCNQ1* looped to the *CDKN1C* promoter as well as to the *INS/IGF2* locus 700 kb distal (Supplementary Fig. 4c), and T2D enhancer variants at the 10q22 locus near *ZMIZ1* looped to the *POLR3A* locus 1 MB distal (Supplementary Fig. 4d). These results define candidate target genes of T2D enhancer signals, including multiple that interact over large genomic distances.

We next mapped candidate target genes regulated by variants at T2D enhancer signals using islet eQTL data. At each signal, we

## Table 1 Candidate genes with eQTLs to T2D enhancer variants

| Locus | # Candidate genes | Enhancer variant[a] | eQTL genes | eQTL P-value[b] | Shared eQTL[c] |
|---|---|---|---|---|---|
| 10p13 | 3 | rs11257655 | CAMK1D | 1.72E-14 | Y |
| 8p11 | 2 | rs508419 | NKX6-3 | 5.59E-10 | Y |
| 12q24 | 13 | rs1260294 | ABCB9 | 2.63E-07 | Y |
| 3q27 | 1 | rs7646518 | IGF2BP2 | 7.49E-07 | Y |
| 2q21 | 3 | rs4954179 | ACMSD | 5.43E-06 | Y |
| 2q21 | 3 | rs4954179 | TMEM163 | 9.69E-05 | Y |
| 15q22 | 6 | rs17205526 | C2CD4B | 0.00088 | Y |
| 4q35 | 1 | rs116401167 | ACSL1 | 0.04 | Y |
| 5q13 | 1 | rs7732130 | PDE8B | 0.048 | Y |

[a]Enhancer variant with highest PPA per signal listed
[b]adjusted eQTL P < 0.05; P-values reported are uncorrected
[c]Bayesian co-localization probability of shared signals is greater than probability of distinct signals

tested the most likely casual enhancer variant for eQTL association to each candidate gene correcting for the total number of candidate genes for that signal (see Methods). For the resulting genes with eQTL evidence (corrected $P < 0.05$), we further confirmed the eQTL and T2D signals did not have distinct causal variants using Bayesian co-localization (see Methods). Target genes showed evidence for islet eQTLs with eight known T2D islet enhancer signals (corrected $P < 0.05$), including CAMK1D, ABCB9, C2CD4B, and IGF2BP2 (Table 1 and Supplementary Table 4). For example, the known T2D variant rs11257655 mapped in an islet active enhancer element that looped directly to the CAMK1D promoter and was an islet eQTL for CAMK1D expression[25] (Fig. 4c). At the 127 putative T2D enhancer signals, we identified 12 additional target genes with evidence for eQTLs to T2D variants (corrected $P < 0.05$) such as FADS1, VEGFA, SNX32, and SCRN2 (Supplementary Table 4). Among these 21 cis-regulated genes, nearly a third have not been identified as significant islet eQTLs in previous studies[7,17,27]. These results identify candidate target genes, which are cis-regulated by T2D islet enhancer signals.

We next characterized the biological functions of candidate genes identified at these T2D enhancer signals. Candidate target genes were strongly enriched in gene sets related to protein transport and secretion, potassium ion transport, vesicles and vesicle membranes, and endoplasmic reticulum (FDR $q < 0.2$) (Fig. 4d and Supplementary Table 5). Candidate target genes also included six genes involved in MODY and other monogenic and syndromic forms of diabetes (ABCC8, KCNJ11, GCK, INS, GLIS3, WFS1) (Supplementary Table 3). Conversely, non-target genes within 1 Mb of these same 30 signals were enriched for gene sets related to stress–response and other processes (FDR $q < 0.2$), which may represent regulatory programs activated in other cellular states (Supplementary Table 5, see Methods). Candidate genes with islet eQTLs to known and putative T2D enhancer signals were specifically enriched for genes involved in vesicle-mediated transport (FDR $q < 0.2$) (Fig. 4d and Supplementary Table 5). These results demonstrate that candidate target genes of T2D enhancer signals are involved in protein transport and secretion pathways.

**Imp2 conditional inactivation affects insulin secretion.** At the 3q27 locus, IGF2BP2 is the only candidate target gene based on T2D variant proximity to the gene promoter and eQTL evidence (Table 1, Supplementary Fig. 5a and Supplementary Table 4), and is furthermore the only gene in the entire TAD (Supplementary Fig. 5a and Supplementary Data 9). We sought to determine the mechanism of risk variant activity in islets at this locus. Fine-

mapped T2D enhancer variants at 3q27 all mapped within a 6 kb intronic region proximal to the IGF2BP2 promoter (Supplementary Fig. 5a and Supplementary Data 7). We tested these candidate enhancer variants for allelic imbalance in islet accessible chromatin (see Methods). We observed significant evidence (FDR $q < 0.1$) for allelic imbalance at rs10428126 (binomial $P = .001$) where the T2D risk (and alternate) allele C had reduced accessibility, and no evidence for imbalance among the other candidate variants at this locus (Supplementary Data 7). This variant has also been reported as a chromatin accessibility QTL in islets[28]. We further validated that the T2D risk allele at rs10428126 reduced islet enhancer activity using gene reporter assays in MIN6 cells ($t$-test $P = 1.0 \times 10^{-3}$) (Supplementary Fig. 5b). In addition, rs10428126 mapped in a site consistently active across ATAC-seq samples and in ChIP-seq sites for NKX2.2 and PDX1, and the risk allele disrupted PDX1 and NKX motifs (Supplementary Fig. 5c). These results reveal a likely causal risk variant at IGF2BP2 that reduces chromatin accessibility and enhancer activity in islets.

As T2D risk alleles at the IGF2BP2 locus are correlated with reduced islet chromatin accessibility, enhancer activity and IGF2BP2 expression as well as reduced insulin secretion phenotypes[24], we hypothesized that reduced activity of IGFBP2 would contribute to a diabetic phenotype in islets. We thus determined the effects of reduced IGF2BP2 (Imp2 in mice) on islet function using a mouse model. Imp2 is widely expressed in adult mouse tissues, including fat, muscle, liver, and pancreas[29], and in the pancreas Imp2 expression localized to islets and overlapped insulin (Fig. 5a). We inactivated Imp2 in mouse beta cells by recombining the Imp2flox(f) allele with Cre recombinase driven by the rat insulin 2 promoter (RIP2-Cre) (Supplementary Fig. 6a). Immunoblot analysis of extracts from isolated Imp2ff/RIP2-Cre islets confirmed reduced Imp2 abundance compared to Imp2ff islets (Fig. 5b). Imp2ff/RIP2-Cre mice exhibited no overt phenotype and gained weight similar to Imp2ff controls on both a normal chow (NCD) and high-fat diet (HFD) (Supplementary Fig. 6b).

We assessed the effect of Imp2 deficiency in mouse beta cells on glucose homeostasis. At 10 weeks of age, Imp2ff and Imp2ff/RIP2-Cre mice on NCD exhibited no difference in blood glucose and insulin levels. By contrast, blood insulin and C-peptide levels were reduced in HFD-fed Imp2ff/RIP2-Cre compared to HFD-fed control mice, whereas blood glucose and glucagon levels were similar (Fig. 5c). When challenged with an intraperitoneal glucose injection, HFD-fed, but not NCD-fed, Imp2ff/RIP2-Cre mice exhibited significantly higher glucose and lower insulin levels than Imp2ff mice (Fig. 5d, e). Importantly, this was not due to a difference in insulin sensitivity, as blood glucose levels after an intraperitoneal insulin injection were similar in Imp2ff and Imp2ff/RIP2-Cre mice (Supplementary Fig. 6c). These results indicate that Imp2 deficiency limits the capacity of beta cells to augment insulin secretion in response to increased insulin demand.

## Discussion

In summary, we defined the genomic location and spatial orientation of accessible chromatin in pancreatic islets. We identified putative target genes for thousands of islet distal enhancers, including those that interacted in chromatin loops over 1 Mb distances. We fine-mapped candidate causal variants in islet enhancers at 30 known T2D signals, and identified an average of one enhancer variant per signal with allelic effects on islet chromatin accessibility. Prioritizing target genes of T2D islet enhancer signals using islet chromatin loops and promoter-proximity greatly reduced the number of potential candidates, and through eQTL mapping we then identified target genes cis-

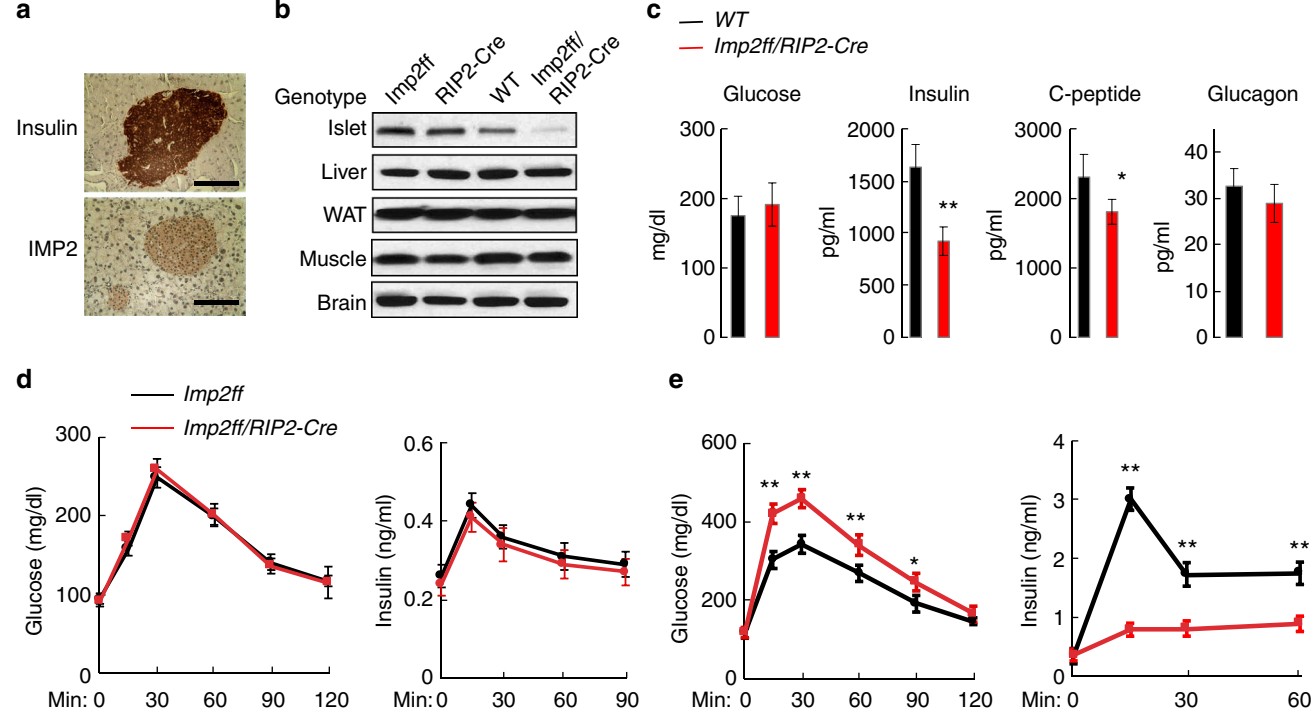

**Fig. 5** Loss of *Imp2* activity in mouse beta cells impairs glucose-stimulated insulin secretion in diet-induced insulin resistance. **a** Immunostaining of insulin and IMP2 in mouse pancreas. Scale bar equals 80 μM. **b** Expression of IMP2 in islets and other T2D-relevant tissues liver, adipose, muscle, and brain. The original uncropped images are provided in the source data file. **c** Blood glucose, insulin, c-peptide, and glucagon level in 10-week-old male mice on high-fat diet (HFD) ($N = 9$). Wild-type (black) and *Imp2ff/RIP2-Cre* (red). **d** One-gram per kilogram of glucose was administered intraperitoneally after overnight fasting of 12-week-old *Imp2ff* (black; $N = 10$) and *Imp2ff/RIP2-Cre* (red; $N = 10$) male mice maintained on normal chow diet (NCD). left = blood glucose; right = serum insulin. **e** One-gram per kilogram of glucose was administered intraperitoneally after overnight fasting to 12-week-old *Imp2ff* (black; $N = 9$) and *Imp2ff/RIP2-Cre* (red; $N = 9$) male mice maintained on HFD. left = blood glucose; right = serum insulin. Values represent mean and SD. *T*-test *$P <$ 0.05, **$P < 0.01$. Source data are provided as a source data file

regulated by T2D enhancer variants. Future studies of chromatin looping generated across larger numbers of samples will enable a greater understanding of risk variants effects on looping directly, as well as correlative relationships with gene expression and other molecular phenotypes. Furthermore, studies modifying islet enhancer activity, for example through genome editing, may provide additional validation of affected target genes in particular those with more subtle effects.

Target genes of T2D islet enhancer signals were specifically enriched in protein transport and secretion pathways, and we validated that reduced activity of *IGF2BP2* homolog *Imp2* in mouse islets leads to defects in glucose-stimulated insulin secretion. The mechanism of how *IGF2BP2* functions in the islet cellular context to produce a diabetic phenotype will be of interest to continued studies. While our results describe gene networks contributing to T2D in normal islet physiology, mapping changes in islet chromatin accessibility and architecture in diabetogenic conditions and disease states will provide additional networks contributing to T2D. These studies will be further enhanced by data from experimental screens for regulatory activity and cellular and animal phenotypes providing more systematic functional validation. Altogether our results link T2D risk to enhancer regulation of protein transport and secretion, and highlight the utility of high-resolution chromatin conformation maps in revealing the genes and networks underlying genetic risk of complex disease.

## Methods

**Islet samples**. Five human islet donors were obtained from the Integrated Islet Distribution Program (IIDP) (Supplementary Table 1). Islet studies had exempt status from the Institutional Review Board (IRB) of the University of California San Diego. Islet preparations were enriched and selected using zinc-dithizone staining.

**Islet ATAC-seq data generation**. We generated ATAC-seq data from four of the human islet samples based on a previously described protocol[12]. For Islet samples ISL_3 and ISL_4, permeabilized nuclei were obtained by resuspending cells in 250 μL Nuclear permeabilization buffer [0.2% IGEPAL-CA630 (I8896, Sigma), 1 mM DTT (D9779, Sigma), Protease inhibitor (05056489001, Roche), 5% BSA (A7906, Sigma) in PBS (10010-23, Thermo Fisher Scientific)], and incubating for 10 min on a rotator at 4 °C. Nuclei were then pelleted by centrifugation for 5 min at 500 x *g* at 4 °C. The pellet was re-suspended in 25 μL ice-cold Tagmentation Buffer [33 mM Tris-acetate (pH = 7.8) (BP-152, Thermo Fisher Scientific), 66 mM K-acetate (P5708, Sigma), 11 mM Mg-acetate (M2545, Sigma), 16% DMF (DX1730, EMD Millipore) in Molecular biology water (46000-CM, Corning)]. An aliquot was then taken and counted by hemocytometer to determine nuclei concentration. Approximately 50,000 nuclei were re-suspended in 20 μL ice-cold Tagmentation Buffer, and incubated with 1 μL Tagmentation enzyme (FC-121-1030; Illumina) at 37 °C for 30 min with shaking 500 rpm. The tagmented DNA was purified using MinElute PCR purification kit (28004, Qiagen). The libraries were amplified using NEBNext High-Fidelity 2x PCR Master Mix (M0541, NEB) with primer extension at 72 °C for 5 min, denaturation at 98 °C for 30 s, followed by eight cycles of denaturation at 98 °C for 10 s, annealing at 63 °C for 30 s and extension at 72 °C for 60 s. Amplified libraries were then purified using MinElute PCR purification kit (28004, Qiagen), and two size selection steps were performed using SPRIselect bead (B23317, Beckman Coulter) at 0.55X and 1.5X bead-to-sample volume rations, respectively. For ISL_1 and ISL_2, frozen nuclear pellets of 50,000 cells were thawed on ice, re-suspended in 50 μL of transposition reaction mix (2.5 μL of Tn5 transposase in 1x TD buffer (Illumina)) for 30 min at 37 °C in a thermomixer with gentle shaking. Immediately following transposition, tagmented DNA was purified using a MinElute Kit (Qiagen) or a DNA Clean and Concentrator-5 kit (Zymo) and eluted in 10 μL of nuclease-free H$_2$O. Five microliters of the purified sample was PCR amplified for 12 cycles using KAPA Real-Time Library amplification kit (KAPA Biosystems) and customized Nextera PCR primers (as in Buenrostro et al.[12]). Amplified libraries were purified using AMPure XP (Beckman Coulter) beads and eluted in 12–15 μL of nuclease-free H$_2$O. Libraries were sequenced on either an Illumina NextSeq 550 or Illumina HiSeq2500.

For each sample, we trimmed adapter sequences using TrimGalore (https://github.com/FelixKrueger/TrimGalore). The resulting sequences were aligned to sex-specific hg19 reference genomes using bwa mem[30,31]. We filtered reads were to retain those in proper pairs and with mapping quality score greater than 30. We then removed duplicate and non-autosomal reads. We called peaks individually for each sample with MACS2[13] at a $q$-value threshold of 0.05 with the following options "—no-model", "—shift -100", "—extsize 200". We removed peaks that overlapped genomic regions blacklisted by the ENCODE consortium and merged the peaks[31]. In total, we obtained 105,734 merged peaks. To assess concordance between ATAC-seq experiments, we calculated read coverage for each peak in the merged set of ATAC-seq peaks, excluding peaks that overlapped blacklisted genomic regions. We then calculated the Spearman correlation between the read counts for each sample.

We collected published ATAC-seq data from primary islets and from FACS-sorted alpha, beta and acinar cells[7,14,15]. We re-processed raw data and called peaks for each sample using the same procedure described above. We determined the overlap in peak calls from the four islet ATAC-seq samples and these published data using Jaccard index.

**Islet Hi-C data generation**. We generated Hi-C data from three of the pancreatic islet samples, two of which also had ATAC-seq data (Supplementary Table 1). In situ Hi-C was performed using a previously published protocol with modifications adapted to frozen human tissue[19]. Briefly, the tissue was cut to fine pieces and washed with cold PBS. Cross-linking was carried out with 1% formaldehyde (sigma) in PBS at room temperature (RT) for 10 min and quenched with 125 mM Glycine (sigma) at RT for 5 min. Nuclei were isolated using a loose-fitting Dounce homogenizer in hypotonic buffer (20 mM Hepes pH 7.9, 10 mM KCl, 1 mM EDTA, 10% Glycerol and 1 mM DTT with additional protease inhibitor (Roche) for 30 strokes and centrifuge at 3500 x $g$ at 4 °C.

Nuclei were digested using 4-base cutter restriction enzyme MboI (NEB) at 37 °C overnight (o/n). Digested ends were filled in blunt with dBTP with biotinylated-14-ATP (Life Technologies) using Klenow DNA polymerase (NEB). Re-ligation was performed in situ when nucleus was intact using T4 DNA ligase (NEB) at 16 °C for 4 h. The cross-linking was reversed at 68 °C o/n while protein was degraded with proteinase K treatment (NEB). DNA was purified with phenol-chloroform extraction and ethanol precipitation, followed by fragmentation to 300–500 bp with the Covaris S220 ultrasonicator. Ligation products were enriched with Dynabeads My One T1 Streptavidin beads (Life Technologies). PCR was used to amplify the enriched DNA for sequencing. HiSeq 4000 sequencer (Illumina) was used to sequence the library with $2 \times 100$ bp paired-end reads.

For each sample, reads from paired-end reads were aligned with bwa mem[32] as single-end reads, and then filtered through following steps. First, only five prime ends were kept for chimeric reads. Second, reads with low mapping quality (<10) were removed. Third, read ends were then manually paired, and PCR duplicates were removed using Picard tools (https://github.com/broadinstitute/picard). Finally, filtered contacts were used to create chromatin contact maps with Juicebox[33].

To determine the consistency in the three Hi-C samples, the contact map from each sample was first binned to 100 kb. Next, each contact matrix was linearized into a single vector. Finally, the Spearman correlation between each pair of samples was measured using the two corresponding linearized vectors. The correlations were calculated using the command scipy.stats.spearmanr in the scipy for python.

Chromatin loops were identified by using HICCUPS[8] at 5, 10, and 25 kb resolutions with default parameters on the Hi-C maps for each individual sample. Next, Jaccard index for loop sets across samples were calculated for overlap within 20 kb using pgltools[34]. The Hi-C data was then pooled across all three samples to create a single contact map, and loops were called with HICCUPs at the same resolutions with the same parameters. A single loop set was then created by unifying the four chromatin loop sets (three from each individual sample, one from pooled data). Specifically, we identified loops where both anchors were within 20 kb of one another via pgltools[34], and retained the loop with the highest resolution; if multiple loops were found at the highest resolution, loops were kept from the contact map with the highest overall sequencing depth. For each resulting chromatin loop in the merged set, we then annotated in which sample(s) the loop was identified in (Supplementary Data 3). We also called topologically associating domains (TADs) from the pooled Hi-C data using a previously described approach[10].

**Islet ChIP-seq data processing**. We obtained previously published data from ChIP-seq assays of H3K4me1, H3K27ac, H3K4me3, H3K36me3, and CTCF generated in primary islets and for which there was matching input sequence from the same sample[4–6]. For each assay and input, we aligned reads to the human genome hg19 using bwa[35] with a flag to trim reads at a quality threshold of <15. We converted the alignments to bam format and sorted the bam files. We then removed duplicate reads, and further filtered reads that had a mapping quality score below 30. Multiple sequence datasets obtained from the same assay in the same sample were then pooled.

We defined chromatin states from ChIP-seq data using ChromHMM[16] with a 9-state model, as calling larger state numbers did not empirically appear to identify additional states. We assigned the resulting states names based on patterns previously described in the NIH Roadmap and ENCODE projects–CTCF (CTCF), Transcribed (Txn; H3K36me3), Active promoter (TssA; H3K4me3, H3K4me1), Flanking promoter (TssFlnk; H3K4me3, H3K4me1, H3K27ac), Weak/Poised Enhancer (EnhWk; H3K4me1), Active Enhancer 1 (EnhA1; H3K27ac, H3K4me1), Active Enhancer 2 (EnhA2; H3K27ac), and two Quiescent states with low signal, which we merged together (Quies/low; low signal for all assays).

We then annotated accessible chromatin sites based on overlap with the chromatin states. If an accessible chromatin site overlapped multiple chromatin states, we split the site into multiple distinct elements.

**Islet chromatin interaction analyses**. To determine the normalized tag counts of ATAC-seq data at loop anchors, loop anchors were converted to a regular BED file with pgltools[34], and HOMER[36] was used to find the normalized tag density across all loop anchors for each ATAC-seq sample. Output from HOMER was normalized to a maximum height of 1 for each sample to determine the distribution of ATAC-seq signal within each sample, rather than the relative magnitude coverage difference between ATAC-seq samples.

To determine the enrichment of each class of islet regulatory elements near loops, and the types of elements co-localized by loops, we utilized pgltools and HOMER to integrate the ATAC-seq and Hi-C data. We first created a size matched null distribution comprised of 7000 permuted regions. Next, for each islet accessible chromatin state, we identified the proportion of states within 25 kb of a loop. We determined the fold enrichment of each class over the average calculated from the null distribution, and determined significance as the number of permuted counts greater than the observed.

To determine which pairs of islet regulatory elements were in chromatin loops at a statistically significant level, we compared the distribution of islet regulatory elements around loop anchors using HOMER. We utilized the "annotateInteractions" function to obtain logistic regression $P$-values and odds ratio enrichment estimates for all pairs of islet regulatory elements.

We defined candidate target genes of islet enhancer elements using Hi-C loops in the following way. First, we identified all islet active enhancer elements mapping directly within a Hi-C loop anchor. We then filtered these loops based on whether the other anchor mapped within a promoter region ($-5$ kb/$+ 2$ kb of transcription start site) for protein-coding or long non-coding genes in GENCODEv27[37]. For each active enhancer, we then calculated the number of gene promoter regions interacting with that enhancer. For each gene promoter region, we calculated the number of independent interactions containing at least one active enhancer element. We also defined a broader set of candidate enhancer and gene promoter interactions by using a 25 kb flanking window around each loop and re-calculating overlap.

We identified genes in direct loops with multiple (>1) active enhancers and tested these genes for gene set enrichment using GSEA[38], considering only gene sets with >25 genes at an FDR of 0.2.

**Genomic enrichment analyses**. We tested for enrichment of variants in each accessible chromatin class using T2D association data of 1000 Genomes project variants from the DIAGRAM consortium[23]. From this meta-analysis, we identified common variants (with minor allele frequency (MAF) > .05). In total, we retained 6.1 M common variants for testing. For each variant, we then calculated a Bayes Factor from effect size estimates and standard errors using the approach of Wakefield[39].

We then modeled the effect of variants in each class of islet regulatory elements on T2D risk using fgwas[22]. For these analyses, we used a window size (-$k$) that resulted in a 1 Mb window on average. We first tested for enrichment of variants in each state individually. We then built a joint model iteratively in the following way. We first identified the annotation with the highest likelihood. We then added annotations to the model until the likelihood did not increase further. Using this model, we introduced a series of penalties from 0 to 0.5 in increments of 0.01 and fit the model using each penalty, and identified the penalty that gave the highest cross-validation likelihood. We then finally removed annotations from the model that further increased the cross-validation likelihood. We considered the resulting set of annotations and effects to be the optimal joint model.

We also modeled the effect of variants in islet regulatory elements using stratified LD-score regression[40]. For these analyses, we extracted variants in HapMap3 from T2D association data. We then calculated LD scores for variants in each regulatory element class. Finally, we obtained enrichment estimates using these LD scores with T2D association data of HapMap3 variants.

**Allelic imbalance analyses**. We collected a total of 23 islet ATAC-seq datasets by combining data from four samples in this study with published data from 19 samples[7,14]. We tested for allelic imbalance at all fine-mapped variants in islet enhancers at 30 known T2D signals, as well as putative T2D variants in islet enhancers outside of known loci. We used bwa mem to align paired-end sequence data, filtered out mitochondrial reads, and removed duplicates using picard MarkDuplicates. For each dataset, we then applied the WASP pipeline to correct for reference mapping bias[27]. To filter out imbalance that could potentially arise from sequencing errors, we set a filter for each sample to limit allelic imbalance testing to heterozygous SNPs with at least two reads covering each allele. At the

*IGF2BP2* locus, we first genotyped a curated set of variants at the locus in each sample using QuASAR with a probability threshold of 0.99[41], and then used the UM Imputation Server to impute genotypes in the HRC reference panel. We then retained variant genotypes with high imputation quality ($r^2 > 0.7$).

For each sample, we then used a binomial test to assess imbalance at each heterozygote assuming a null hypothesis where the two alleles were equally likely to be observed. For each variant, we calculated signed $Z$-scores from $P$-values and combined signed $Z$-scores across samples with Stouffer's $Z$-score method using sequencing depth to weight statistics from each sample. From the resulting combined $Z$-scores, we calculated combined $P$-values and $q$-values using the Benjamini-Hochberg procedure. The $q$-values were calculated separately for known T2D signals (using variants in Supplementary Data 7) and the putative T2D signals (using variants in Supplementary Data 8).

For the set of variants at known and putative T2D signals tested for allelic imbalance in islet ATAC-seq signal, we further tested these variants for allelic imbalance in islet Hi-C. We used samtools mpileup to obtain allele counts for each variant in each sample[42], retained variants with at least three reads covering each allele, and calculated a $P$-value from the resulting counts using a binomial test. We then combined $P$-values for each variant across samples using Fisher's meta-analysis. For variants at known loci with significant ATAC-seq imbalance, we determined the number of variants that had the same direction of effect on allelic imbalance in islet ATAC-seq and islet Hi-C signal. If there were multiple variants at a given signal with ATAC-seq imbalance, we retained only the variant with the highest PPA. We then used a binomial test to assess whether there was significant directional concordance in ATAC-seq and Hi-C imbalance under an expected concordance of 50%.

**Genomic annotation analyses for TF ChIP-seq.** We obtained 13 published islet transcription factor ChIP-seq datasets[4,43] for FOXA2 (2), MAFB (2), NKX2-2 (2), NKX6-1 (2), and PDX1 (5) and re-processed each experiment with a uniform processing pipeline. We used bwa aln and bwa samse to align the reads to hg19 with a quality threshold of 15. We then removed duplicate reads with the picard MarkDuplicates tool. We called peaks with MACS2[13] using matched input controls for each experiment after extending reads to a uniform 200 bp length.

**Genomic annotation analyses for DNA methylation.** We obtained low-methylated regions (LMRs) and un-methylated regions (UMRs) defined in a published islet DNA methylation study[14]. In this study, LMRs were defined as having methylation ranging from 10–50% and containing fewer than 30 CpG sites.

**Genomic annotation analyses for TF footprints.** To identify haplotype-aware motifs within ATAC-seq footprints overlapping accessible chromatin sites, we searched accessible chromatin sites from four ATAC-seq samples for instances of motifs from JASPAR, SELEX, ENCODE, and de novo motifs identified in our data. We used vcf2diploid[44] (https://github.com/abyzovlab/vcf2diploid) to create individual-specific diploid genomes by mapping our phased, imputed genotypes onto hg19 using only SNPs and ignoring indels. Then, we used fimo[45] to scan the personalized genomes for our compiled database of motifs, limiting the sequences scanned to those derived from islet accessible chromatin. For fimo, we used the default parameters for $P$-value threshold ($1 \times 10^{-4}$) and a background GC content of 40.9% based on hg19.

Using make_cut_matrix from atactk v 0.1.6 (https://github.com/ParkerLab/atactk) we created cut-site matrices containing the number of Tn5 integrations ± 100 bp around each sequence motif on both the forward and reverse strands. We then determined the posterior probability that a given motif occurrence was bound using CENTIPEDE[46], and defined footprints using a posterior probability threshold of 0.99. We merged the resulting footprints with a published set of pancreatic islet ATAC-seq footprints[7].

We further identified variants predicted to disrupt each footprint. We calculated the entropy score for a variant position in a footprint using the position frequency matrix (PFM) for each motif. A footprint was considered disrupted if a variant fell in a conserved position in the motif (defined as entropy <1.0).

**Fine-mapping of T2D risk variants.** We used the effects from the joint enrichment model as priors on the evidence for variants at 107 known T2D signals using fine-mapping data from the Metabochip[2], GoT2D[1] and DIAGRAM 1000 Genomes[23] studies. We used data of 49 T2D signals at 39 T2D loci on the Metabochip, 41 additional T2D signals from GoT2D data for T2D loci not on the Metabochip, and 17 additional T2D signals in DIAGRAM 1000G not in Metabochip or GoT2D.

For each signal, we obtained the enrichment effect of the islet regulatory or coding annotation overlapping that variant. We calculated a prior probability for the variant by dividing the effect by the sum of effects across all variants at a signal. We then multiplied this prior probability by the Bayes Factor for each variant. From the resulting odds, we calculated a posterior probability that the variant is causal for T2D risk (PPA) by dividing the odds by the sum of odds across all variants at the locus.

For each signal, we calculated a cumulative PPA (cPPA) value for islet enhancer (EnhA1, EnhA2, EnhWk), promoter (TssA, TssFlnk), CTCF-binding site, UTR, and coding exon (CDS) annotations by summing the PPAs of all variants

overlapping each annotation. We then clustered T2D signals into groups based on cPPA values using $k$-means clustering.

We determined the effects of T2D signals in each cluster on glycemic association data[24]. We identified 73 T2D signals represented in these data and cataloged 23 associated at adjusted $P < 0.05$ with first-phase insulin response, peak insulin response, AIR, or insulin secretion rate. We calculated the percentage of signals in each cluster associated with these measures and tested for differences between clusters using a Chi-square test with a $2 \times 2$ contingency table segregated by (1) enhancer signal, or un-annotated signal and (2) associated with insulin secretion measures, or not associated with insulin secretion measures.

We also performed a binomial test of the number of enhancer signals associated with insulin secretion measures using the fraction of un-annotated signals associated with insulin secretion measures as the expected value and found a similar enrichment ($P = 2.4 \times 10^{-3}$).

For the 30 T2D islet enhancer signals, we calculated 99% credible sets as the set of candidate variants that explain 99% of the total PPA using genetic fine-mapping data alone (genetic), and fine-mapping, including priors from the joint genome-wide enrichment model (+priors).

We then fine-mapped casual variants in putative T2D loci genome-wide. For variants in each 1 MB window across the genome, after excluding any windows overlapping a known T2D signal, we obtained the effect of the islet annotation overlapping that variant. We calculated a prior probability for each variant as described above also including an additional prior on the evidence that the 1 MB window is a T2D locus. We multiplied both prior probabilities by the Bayes Factor for each variant. From the resulting odds, we calculated the PPA that each variant is causal for T2D risk. We then considered the 131 windows with at least one islet enhancer variant with PPA > 0.01 in downstream analyses.

**Genomic features analyses.** For each class of islet open chromatin, we determined the overlap with other genomic features.

We identified motifs enriched in sequence underneath each islet accessible chromatin class. We first extracted genomic sequence for each site using bedtools[47], and masked repetitive sequences. We then identified de novo motifs enriched in this sequence using DREME[48]. For each de novo motif, we determined whether this motif matched a known sequence motif in a custom database of >2500 motifs from ENCODE, JASPAR, and SELEX with tomtom[31,49–51].

We determined the overlap of islet accessible chromatin classes with transcription factor (TF) ChIP-seq data in islets for five proteins[4,31]. For each islet chromatin class, we calculated the Jaccard index of overlap with sites for each TF[47]. We then determined the overlap of islet accessible chromatin classes with DHS sites from 384 cell types in the ENCODE project[31]. We first filtered out DHS sites from islets, and then for each accessible chromatin site, we calculated the percentage of ENCODE cell-types the site was active in. We then determined the median percent overlap across all sites within each accessible chromatin class.

**Gene expression analysis.** We obtained transcript-per-million (TPM) counts from RNA-seq data in 53 tissues from the GTEx project release v7[21]. We further obtained RPKM read counts from RNA-seq data of 118 pancreatic islet samples[20], and converted RPKM to TPM values using the formula $TPM_i = (RPKM_i/sum (RPKM_j) \times 10^6$ for gene i and sample j[52]. We then retained only protein-coding and long non-coding genes annotated in GENCODEv27[37]. We calculated the number of genes expressed in islets (defined as ln(TPM) > 1) and not expressed in islets with and without at least one reference active enhancer chromatin loop to the promoter region. We then tested for a significant difference using a Chi-square test on a $2 \times 2$ contingency table of genes (1) expressed, or not expressed in islets and (2) at least one enhancer loop, or no enhancer loop.

Across all 54 tissues, we filtered out genes not expressed (ln(TPM) > 1) in at least one tissue. We determined correlation between gene expression level in islets and enhancer loop number using Spearman's rho. We annotated each gene with the number of loops to enhancers in our reference set, and then determined the correlation between the expression level of the filtered genes with the number of reference enhancer loops annotated to the gene. We further grouped genes by the number of chromatin loops to enhancer elements and calculated the median islet TPM value for each group.

We then determined the relative expression level for each gene in 54 tissues. We quantile normalized expression values within each tissue, log-transformed the resulting values and then calculated a $Z$-score for each gene using the mean and standard deviation across tissues. We then repeated the above analyses using tissue $Z$-scores instead of tissue TPM values. We further created a linear model of gene $Z$-scores with chromatin loop number as the predictor using the glm package in R. Values are reported as the effect size (beta) and standard error from the resulting model.

**Islet expression QTL analysis.** We obtained summary statistic eQTL data from two published studies of 118 and 112 primary pancreatic islet samples[20,53]. We then performed inverse sample-size weighted meta-analysis to combine the summary results for each variant and gene pair using METAL[54]. We retained only protein-coding and long non-coding RNA genes as defined by GENCODEv27[37],

only variant and gene pairs tested in both studies, and only variants with minor allele frequency (MAF) > 0.01.

We extracted eQTL associations for variants in classes of islet accessible chromatin. To remove potential biases due to linkage disequilibrium, we sorted variant associations based on $P$-value and iteratively pruned out variants in LD ($r^2$ > 0.5) with a more significant variant using LD information in European samples from 1000 Genomes project data. We then extracted pruned eQTL associations for variants in active promoter elements for genes within 20 kb (TssA), variants in active enhancer elements for genes within 20 kb (Enh. proximal), variants in active enhancer elements for genes in chromatin loops (Enh. loop), and variants in active enhancer elements for genes without a loop (Enh. no-loop). For each set of eQTL associations, we compared $P$-value distributions using a two-sided Wilcox rank-sum test. To remove biases in variant distances to loop and no-loop genes, we randomly selected variant-gene pairs matched on distance to the distal target set (Enh. no-loop matched) and re-performed analyses. We also performed these analyses using enhancer and gene promoter pairs within 25 kb of the loop boundaries.

**Target genes of T2D islet enhancer signals**. We defined candidate target genes of 30 known T2D enhancer signals and 127 putative T2D enhancer windows in the following way. We identified candidate causal variants at each signal overlapping islet enhancer elements and considered target genes as those where (a) the enhancer and promoter region were within 25 kb of a chromatin loop or (b) the enhancer was within 25 kb proximal to the promoter region.

We next defined alternate sets of target genes of the 30 T2D enhancer signals based on 1 MB windows or TAD boundaries. For 1 MB window definitions, we identified the highest probability variant for each signal and extracted a $+/-1$ MB window around the variant position. We then considered gene promoter regions for protein-coding or long non-coding genes in GENCODEv27 that overlapped this $+/-1$ MB window the set of target genes. For TAD boundary definitions, we intersected the merged set of TADs with gene promoter regions to obtain a set of genes within each TAD. We then intersected the highest probability variant at each T2D signal with TADs to obtain gene sets in the TAD.

For each enhancer signal with a candidate target gene, we extracted eQTL $P$-values for each target gene using the islet enhancer variant with the highest PPA at the signal. Where the highest probability variant was not present in the eQTL dataset, we used the next most probable islet enhancer variant. We then Bonferroni-corrected eQTL $P$-values for the total number of candidate target genes at the signal and considered eQTLs significant with a corrected $P$-value < 0.05.

For genes with significant eQTL evidence we further tested whether T2D and eQTL signals were co-localized. We obtained the T2D Bayes Factor for each variant at the signal from fine-mapping data. For significant gene eQTLs at the signal, we then calculated the Bayes Factor that each variant is an islet eQTL for that gene[39]. We compared Bayes Factors for T2D signals and eQTLs for each gene using Bayesian co-localization[55]. We considered the prior probability that a variant was causal for T2D risk or an islet eQTL as $1 \times 10^{-4}$, and the prior probability that a variant was causal for both T2D risk and an islet eQTL as $1 \times 10^{-5}$. We considered signals as having evidence for co-localization where the probability of a shared causal variant was higher than the probability of two distinct causal variants.

We tested target genes for gene set enrichment using GSEA[38], considering only gene sets with >25 total genes and at an FDR threshold of 0.2.

**Luciferase reporter assays**. To test for allelic differences in enhancer activity we cloned sequences containing alternative or reference alleles of tested variants upstream of the minimal promoter of firefly luciferase vector pGL4.23 (Promega) using *Kpn*I and *Sac*I restriction sites.

The primer sequences were:
rs7732130
Forward/left: GATAACGGTACCGCGAAGTGGTCATGGGTAAA
Forward/right: AAGTAGGAGCTCACCATCCCAGCATTTAGTGG
Reverse/left: GATAACGAGCTCGCGAAGTGGTCATGGGTAAA
Reverse/right: AAGTAGGGTACCACCATCCCAGCATTTAGTGG
rs10428126
Forward/left: GATCTCGAGCTCTTCATGAATGCAGGGACAGA
Forward/right: GGTACCGGTACCGCTGCATTGGGTTTTGAAAT

MIN6 beta cells were seeded into 6 (or 12)-well trays at one million cells per well. At 80% confluency, cells were co-transfected with 400 ng of the experimental firefly luciferase vector pGL4.23 containing the alt or ref allele in either orientation or an empty vector and 50 ng of the vector pRL-SV40 (Promega) using the Lipofectamine 3000 reagent. All transfections were done in triplicate. Cells were lysed 48 h after transfection and assayed for Firefly and Renilla luciferase activities using the Dual-Luciferase Reporter system (Promega). Firefly activity was normalized to Renilla activity and compared to the empty vector and normalized results were expressed as fold change compared to empty vector control per allele. Error bars are reported as standard deviation. A two-sided $t$-test was used to compare luciferase activity between the two alleles for a given orientation. MIN6 cells were obtained from the Jhala lab, University of California San Diego.

**Mouse *Imp2* targeting construct and physiological studies**. We generated the Imp2 construct by using a genomic fragment of 12 kb containing *Imp2* exons 1 and 2, as well as flanking intron sequences of the murine gene extracted from the RP23-163F16 BAC clone. The replacement-type targeting construct consisted of 9.4 kb of Imp2 genomic sequences (4.4 kb in the left homology arm and 5.4 kb in the right homology arm) (Supplementary Fig. 6a).

We bred mice for experiments by crossing IMP2-loxp mice (*Imp2*ff) with RIP2-Cre mice on a C57Bl/6J background. We maintained colonies in a specific pathogen-free facility with a 12:12 light–dark cycle and fed irradiated chow (Prolab 5P75 Isopro 3000; 5% crude fat; PMI nutrition international) or a HFD (D12492i; 60% kcal fat; Research Diets Inc.). Blood glucose, insulin, C-peptide and glucagon levels were measured by the Vanderbilt metabolic core. Measurements for *Imp2*ff and *Imp2*ff/*RIP2-Cre* mice were performed using male mice under basal conditions ($N = 10$), upon intraperitoneal glucose injection ($N = 9$), and upon intraperitoneal insulin injection ($N = 9$). A two-sided $t$-test was used to compare differences in measurements across genotypes.

All animal procedures were approved by the Institutional Animal Care and Use Committee of Massachusetts General Hospital and were performed in accordance with the NIH principles of laboratory animal care.

## Data availability

The data in this study are available under accession numbers PRJN527099, TSTSR043623, and TSTSR081148. The source data underlying Figs. 2a, c–f, 3f, h, 4a, 5b–e, Supplementary Fig. 1a, Supplementary Figs. 1c, 2f–h, 3c, 5b, 6b, c are in the Source Data File; other data for figures are in supplementary tables and data. All other data are contained within the article and its supplementary information or upon reasonable request from the corresponding author.

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

## Acknowledgements

Support for this work was provided by NIH funding U01DK105541 to M.S., K.A.F. and B.R., R01DK114650 to K.G., U01DK112155 to M.S. and K.A.F. and R37DK017776 and P30DK057521 to J.A., the Ludwig Institute for Cancer Research to B.R., postdoc fellowship 537-2014-6796 from the Swedish Vetenskapsrådet to J.Y., fellowship F31HL142151 to W.W.G. and JDRF-3-2012-177 postdoc fellowship to A.W., D.K. and L. M. thank the EMBL-Monterotondo gene expression and transgenic services for the production of the Imp2 floxed line. L.M. was supported in part by the Lundbeck foundation (177/05) and EMBL-Mouse Biology Unit. D.K. was supported by an EMBL-postdoctoral fellowship. Funding for mouse experiments was provided by the Slim Initiative for Genomic Medicine, a project funded by the Carlos Slim Foundation in Mexico.

## Author contributions

K.J.G., B.R., M.S., K.F. conceived of and supervised the research in the study; K.J.G. wrote the manuscript and performed analyses; W.W.G, J.C., Y.Q. performed analyses and contributed to writing; J.Y. performed Hi-C assays and contributed to writing; N.D., J.A. performed mouse experiments and contributed to writing; A.W., A.A. contributed to analyses and data interpretation; J.Y.H., N.V., F.D., D.G. performed ATAC-seq assays and contributed to data interpretation; N.K. and M.O. performed variant reporter experiments; L.R., D.K. and L.M. contributed to mouse experiments.

## Additional information

**Competing interests:** The authors declare no competing interests.

