## [Peer Review File · Nature Communications]

Reviewer #2 (Remarks to the Author):

Greenwald et al have responded to the majority of my comments. However, some key issues remain. I agree that it is important to assign genes to enhancers and the use of HiC is important. However, the number of samples used in this study for HiC is very small and a larger number of samples is needed to make the data representative for humans, including different genotypes, and it would also be much better to include both donors with type 2 diabetes and controls. For example, only 40% of loops they present and use for analysis were identified in more than one sample. I appreciate that this represents the first high resolution 3D chromatin map of islets. However, I believe it should be done in a larger number of islet donors to merit publication in Nature Communications. In particular, since the data are used to study genetics, one would like donors representing each genotype for the studied variants.

In their response to comment 1, they write “We agree with the reviewer that multiple studies have demonstrated the importance of islet enhancers in genetic risk of T2D. In fact, this was the primary motivating factor behind the present study - the target genes of islet enhancer activity are largely unknown, which is necessary to understand how enhancers influence T2D risk.”. However, to understand how enhancers influence T2D, they would need to perform HiC in larger number of samples including subjects with different genotypes for the T2D risk SNPs not just perform fine mapping and/or perform these studies in enough samples from donors with T2D and controls.

Regarding their response to comment 2: On page 6 line 170-171, I believe they added the wrong references? Should references for eQTLs be included instead?

Regarding their response to comment 3-4:, they have now added references to additional studies which have used ATAC-seq in human islets and in sorted alpha and beta cells. They also correlated their ATAC-seq data with published data and present these correlations in Figure S1B.

They use merged ATAC-seq peaks for 4 samples in their analyses (n=105,734). In Table S1, they now present the number of reads and peaks per sample. There is a large variation in the number of reads (17,225,636-73,136,422 reads) and peaks (45,453-76,833 peaks) for the samples where the number of reads clearly affect the number of identified peaks per sample. I do not believe that this approach (where peaks are merged when the technical variation is very large) is very good. This means that if a peak is present in one person it is used for analysis. Addition of ATAC-seq data from other groups improve their results.

Additionally, published ATAC-seq data were used for allelic imbalance and to test if T2D SNPs are located in open chromatin regions (Table S9). In Table S9, one problem is that the majority IGF2BP2

SNPs presented do not seem to be located in open chromatin regions based on majority of ATAC-seq data and do not seem to have any allelic imbalance (except rs10428126, which seems to give allelic imbalance but is only located in ATAC-seq peaks/open chromatin region of 7 of 23 donors)?

Regarding their response to comment 5: I do not find any definition of low methylation regions (LMRs) in the paper? What absolute percentage methylation does it represent?

Regarding their response to comment 6: I appreciate that they have toned down their conclusions. The functional IGF2BP2 data are interesting. But the IGF2BP2 ATAC-seq data in Table S9, are not very supportive (see comment 3-4 above).

Regarding their response to comment 7: Could you please include number of aligned reads and number ATAC-seq peaks generated for frozen and fresh samples for ISL3, as well overlapping peaks as not just a correlation.

Regarding their response to comment 15: I am not convinced about the method/analysis used here, where they correlated the number of loops annotated to a gene using their HiC data set of 3 samples with average level of expression in 118 islets samples in a different data set. Gene expression of their own samples should be used here and individual samples should be used for correlation analysis between HiC and expression data in a bigger number of samples.

Reviewer #3 (Remarks to the Author):

In this study Hi-C data is generate from on human pancreatic islets, and this data is used to better determine the promoters distal enhancers actually associate with and thus which target genes variant containing enhancers might actually be regulating. This is important and valuable insight. By combining this data with eQTL data they identify 8 enhancers that interacted with genes with T2D eQTLs, including CAMK1D, ABCB9, C2CD4B, and IGF2BP2.

They then focus on IGF2BP2, and knock-out this gene in mouse beta-cells using the RIP2-Cre driver. They find IGF2BP2 (Imp2 in mice) deficient mice have HFD diet dependent glucose intolerance.

This revised version of the manuscript address my previous concerns and seems to largely address the concerns of the other previous reviewer. I know think it is suitable for publication.

Reviewer #4 (Remarks to the Author):

Comments on Greenwald et al: Pancreatic islet chromatin accessibility and conformation defines distal enhancer networks of type 2 diabetes risk (180095 Nature Communications)

Comments for the authors:

The authors perform ATAC-seq and Hi-C, and in combination with published CHIP-seq, GWAS and eQTL data use this to map putative regulatory elements in pancreatic islet cells, and to link these regulatory regions to potential target genes. The authors thereby generate a catalogue of regulatory elements and their putative target genes in pancreatic islets, which they characterize using chromatin state analyses and transcriptome data. Target genes were found to be enriched in genes involved in protein transport and secretion pathways. The effect of two risk allele variants on transcription is assayed in reporter gene assays. Finally, the authors delete one target gene of a type 2 diabetes (T2D) risk allele, the *Imp2* gene (*IGF2BP2* in human) in mice, and show that the absence of this gene results in defects in glucose-stimulated insulin secretion.

The experimental approach of linking disease-associated sequence variants to target genes using chromosome conformation capture techniques is timely, and highly relevant to uncover regulatory mechanisms that underlie disease, as in this case T2D. This is especially true in the light of the discovery that the vast majority of GWAS hits map to regions with regulatory potential located in the non-coding genome (Maurano et al., 2012). The experiments in this study are conducted and controlled adequately. However, unfortunately the manuscript contains a series of mistakes which add up to the impression of a hastily put together and as a consequence, very difficult to read manuscript. Some of the findings presented merely confirm previous results (for example the finding that gene expression levels correlate positively with the number of interacting enhancers, which has been shown before in several papers using Promoter Capture Hi-C). I also have concerns about the very high number of chromatin loops that involve promoters and enhancers; this is in disagreement with previously published studies (see below). In conclusion, the manuscript in its current form is not suitable for publication, and I am not convinced that the novelty of the findings presented here justifies publication in Nature Communications. I would therefore advise against publication.

Major points:

1.) Introduction, lines 73-75: "...we generated the first high-resolution, genome-wide map of 3D chromatin architecture in pancreatic islets, and used this map to annotate islet enhancers defined using ATAC-seq assays and published ChIP-seq data."

This is not correct. The 3D chromatin architecture map is not used to annotate enhancers - this is done using ATAC-seq and ChIP-seq. The 3D chromatin architecture map is used to link these annotated enhancers to their putative target genes.

2.) In the methods (line 443), the CMM states are explained as "...Active Enhancer 1 (EnhA-1; H3K27ac), Active Enhancer 2 (EnhA-2; H3K27ac, H3K4me1)". This does not match with what is shown in Figure S1C. The enrichment for H3K4me1 is stronger for EnhA-1 than for EnhA-2, both in absolute terms, and in comparison to H3K27ac within the respective enhancer categories.

3.) Figure S1C: is the chromatin state 'Quiesc' duplicated here on the y axis? This state seems to be devoid of any chromatin on the x axis. Why is it called 'quiescent', which is a rather unusual description for a chromatin state (quiescence in its usual definition refers to a state in which cells do not divide but retain the ability to re-enter cell proliferation). Why not call this state 'background' or 'no mark'?

4.) Are figures S1D, S1E and S1F mentioned in the text? If not, why are they there?

5.) Lines 141-142: "Conversely, the promoter regions of 8,448 genes had at least one loop to an enhancer element."

This seems extremely high. How many loops are there that do not involve a promoter or enhancer, out of the 11,924 loops identified in total? Most loops in deep-sequenced Hi-C data appear to be between CTCF sites (for example Rao et al., Cell 2014), and only a minority involve promoters or enhancers. How do the authors explain this discrepancy to published data?

6.) Figure 2E is confusing. First, the proximal and Enh Distal (target) categories are referred to as dark blue and blue, respectively - these look the same shade of blue to me. Second, two of the categories on the x axis are called the same 'Enh Distal (no target)'. My understanding is that both describe genes not involved in loops, but one is distance-matched (the one referred to as light blue; maybe 'turquoise' is better?) - this needs to be pointed out in the figure itself.

7.) Lines 293-295: "We further validated that the risk allele at rs10428126 reduced islet enhancer activity using gene reporter assays..."

In this case (Fig. S5A), as in Figure 3H, the Ref alleles drive higher reporter gene expression. However, here, the authors conclude that "that the risk allele at rs10428126 reduced islet enhancer activity", whereas "...the risk allele at rs7732130 increased enhancer activity..." (line 241). Are both these statements correct?

Better labelling of the figures would help to avoid confusion. Is the Ref allele the respective risk allele in both Figs 3H and S5A?

8.) The *Imp2* inactivation experiment addresses the role of IMP2/IGF2BP2 in glucose metabolism, but I cannot see how this is directly linked to an enhancer risk variant? The gene deletion will no doubt result in more dramatic downregulation of gene activity than a sequence variant in the enhancer?

9.) What is shown in Figure 5D and E? According to the figure legends for 5D and 5E, pretty much the same thing?

Lines 879-881: "(D) 1 g/kg glucose was administered intraperitoneally after overnight fasting of 12-week-old *Imp2ff* (black; N=10) and *imp2ff/RIP2-Cre* (red; N=10) male mice maintained on normal chow diet (NCD)."

Lines 882-884: "(E) 1 g/kg glucose was administered intraperitoneally after overnight fasting to 12-week-old *Imp2ff* (black; N=9) and *Imp2ff/RIP2-Cre* (red; N=9) male mice maintained on NCD."

By contrast, lines 315-318 in the main text state: "When challenged with an intraperitoneal glucose injection, HFD-fed, but not NCD-fed, *Imp2ff/RIP2-Cre* mice exhibited significantly higher glucose and lower insulin levels than *Imp2ff* mice (Figure 5D,E).

I suspect (D) shows data from NCD-fed mice, and (E) shows data from HFD-fed mice?

Minor points:

10.) Affiliation: 13. Present affiliation: Center for Epigenomics, UC San Diego, La Jolla CA - there is no 13 associated with any of the authors from what I can see. Also, is this not identical to 7 (Center for Epigenomics, UC San Diego, La Jolla CA)?

11.) Abstract, lines 43-45: "We identified target genes for thousands of distal islet enhancers, many interacting over 1Mb distances, and genes interacting with enhancers were correlated with islet-specific expression patterns."

An example where more precise language would help, see also points 12 and 13 below. Should this be: "We identified target genes for thousands of distal islet enhancers, many interacting over 1Mb distances, and found that genes interacting with enhancers were correlated with islet-specific expression patterns."

12.) Abstract, lines 48-50: "We defined target genes of these T2D islet enhancer signals using chromatin looping and islet eQTL mapping, and target genes were specifically enriched in protein transport and secretion pathways."

Should this be: "We defined target genes of these T2D islet enhancer signals using chromatin looping and islet eQTL mapping, and found that target genes were specifically enriched in protein transport and secretion pathways."

13.) Introduction, lines 62-63: "The genes regulated by islet enhancers are largely unknown, however, impeding the discovery of disease-relevant gene networks and pathways perturbed by risk variants and novel therapeutic pathways."

As it stands this reads as "...novel pathways perturbed by risk variants."

Better would be, for example: "The genes regulated by islet enhancers are largely unknown, however, impeding the discovery of disease-relevant gene networks and pathways perturbed by risk variants and the development of novel therapeutic pathways."

14.) Figure S1B: ISL1_frozen, ISL2_frozen, ISL3_frozen, ISL4_frozen: do these correspond to ISL1 to ISL_4 in Figure S1A? There is an additional sample ISL3_fresh in Figure S1B?

15.) Results, lines 94 to 96: "..., as well as strong concordance in peak calls from published data of ATAC-seq from 19 islet samples and FACS-sorted beta and alpha cells."

Is such a general statement justified here? The concordance appears to stretch over a wide range of values, and I don't see strong concordance in every example. For example, the concordance between ISL3_frozen/ISL_fresh3/ISL4_frozen with Acinar_2/Beta2/HP1443Hg19 is rather weak.

16.) Figure 1A: can the authors explain why accessible chromatin is much more enriched in the EnhA1 state compared to EnhA2? The authors statement that "Accessible chromatin predominantly mapped with active enhancer and promoter states" (line 99) is true for EnhA1 and TssA, but not for EnhA2, which is less enriched than the Quies state for ATAC-seq signals.

17.) Lines 104 - 105: "...preferentially harboured motifs for FOXA, RFX and NEUROD and other islet transcription factors (Figure S1, Table S3).

Two issues here:

I assume this refers to Figure S1F? And why do the authors not show the data for RFX and NEUROD in this supplemental figure, if those are indeed the most enriched motifs? In Figure S1F, the enrichment for NKX6.1, FOXA2, MAFB, PDX1 and NKX2.2 are shown.

18.) Figure legend S1E (lines 896-897): "islet regulatory elements" - these should be called chromatin states. I can't see any justification for calling the 'Quies' state a regulatory element.

19.) Figure legend S1F (line 898): "islet regulatory elements" - these should be called chromatin states. I can't see any justification for calling the 'Quies' state a regulatory element, see point 18 above.

20.) Figure S1F: the Jaccard metric is missing (whereas it is present in for example S1B).

21.) Lines 125-126: "Nearly half of all islet regulatory elements were proximal to an anchor..."

See points 18 and 19 above. I don't think 'regulatory elements' is correct here. The authors refer to chromatin states, which include one category (quiescent) which is devoid of any of the assayed chromatin marks and thus cannot be assigned regulatory element status by any criteria that I can think of.

22.) Line 147-148: "..., there were four distinct loops between active enhancers and the MAFB promoter, including several loops to enhancers over 1 Mb distal."

Several in this case means two. Just say two.

23.) Lines 828-829: "Multiple islet enhancers formed chromatin loops with the MAFB promoter region including several over 1MB."

Related to point 22 above. It's four enhancers, and two interactions over 1Mb. Just state it as it is: "Four islet enhancers formed chromatin loops with the MAFB promoter region including two over 1MB."

24.) Figure 2B: which chromosome is shown? What is the colour code for the chromatin states? As for the chromatin state colours, are they the same as in Figure 1D? If yes, how will the readers be able to distinguish between EnhA1 and EnhA2, which seem to be the same shade of dark blue?

25.) Figure 2B right panel: although difficult to assess at this resolution, at least significant loops appear very close to the diagonal, whereas other clearly visible interactions in this heatmap have not been called as significant. Are the authors confident in their loop-calling, especially in the light that the enrichment of promoters and enhancers at their chromatin loop anchors differs markedly from other reports (Rao et al., 2014), see point 5 above?

26.) Lines 116 - 118: "We merged the resulting four sets of loop calls where both anchors overlapped at 20 kb resolution." Do the authors think this kind of resolution is sufficient to analyse individual enhancer-promoter interactions, for example. A competing study in bioarchive (Miguel-Escalada et al.) uses Promoter Capture Hi-C which offers single restriction fragment resolution...

27.) Lines 220-221: "Outside of known loci, we identified an additional 131 1Mb windows genome-wide harbouring putative T2D variants in islet enhancers."

How can the authors pinpoint variants within a 1 Mb window to specific enhancers?

28.) Figure 3H: abbreviations need to be explained in the figure legend (Alt = alternative; Ref = reference). Which one is the risk allele? According to the text, it would have to be the Ref allele, as this is the one driving increased luciferase activity? See also point 7 above.

29.) Lines 185-187: "The effects of variants in regulatory elements in T2D risk in the context of chromatin looping, however, is unknown."

Should be: "The effects of variants in regulatory elements in T2D risk in the context of chromatin looping, however, are unknown."

30.) Line 216-217: "...and at 6 signals resolved a single causal enhancer variant such as at ZBED3 (Figure 3G...)."

How was this enhancer variant linked with ZBED3? Using eQTL data? Chromatin looping data? Both? The enhancer seems to be located equidistant to ZBED3 and PDE8B.

31.) The genomic coordinates are missing for Figure 3G.

32.) Lines 259-260: "...; for example, multiple KCNQ1 signals interacted with INS/IGF2 over 700 kb distal,..."

What exactly do the authors mean by 'KCNQ1 signals'? Sequence variants located in putative enhancers downstream of KCNQ1?

33.) Lines 260 - 261: "..., and ZMIZ1 interacted with POLR3A over 1MB distal." Again, the authors should use more precise language here. Are these sequence variants in introns of ZMIZ1?

34.) Figure 4E: no chromosome number or sequence coordinates are provided.

35.) Lines 284-285; "..., and IGF2BP2 is the only implicated target gene at its respective locus in our analyses."

How has IGF2BP2 been implicated? On eQTL data and Hi-C data? Does the table in Figure 4D refer to Hi-C chromatin looping data (column '# target genes')? How far away is rs10428126 from the IGF2BP2

promoter? Is the Hi-C data resolution sufficient to link this sequence variant to the IGFBP2 promoter?

36.) Regrettably, the authors have not attempted to find a uniform font type or size for the figures.

We thank the reviewers for their helpful comments. In response to comments from Reviewer #2 and #4 we have made revisions to the manuscript text and analyses that fully address these comments, and we hope the reviewers agree that these additional revisions have resulted in an improved study that is suitable for publication.

To summarize the changes in this revision we have:

- **Added additional Hi-C allelic imbalance data which now covers the majority (60%) of fine-mapped T2D enhancer variants at known and putative loci.**
- **Demonstrated a high degree of reproducibility in islet accessible chromatin sites across samples and other islet chromatin data, and revised the manuscript to focus on fine-mapped T2D risk variants in reproducible sites (identified in >1 ATAC-seq sample).**
- **Clarified fine-mapping results at the *IGF2BP2* signal which show that rs10428126 is the likely causal variant at this signal across multiple lines of evidence including (i) overlap with a reproducible islet accessible chromatin site, including in data from additional ATAC-seq samples, (ii) gene reporter data demonstrating that the site is a functional enhancer in beta cells, and (iii) allelic imbalance in islet accessible chromatin and allelic effects on enhancer activity in gene reporters, and which is further validated in an independent study.**
- **Substantially revised the analyses of enhancer and promoter looping to describe sites in direct chromatin loops. In these results we identify 3,022 islet enhancers in a direct loop to a gene promoter, and 2,028 gene promoters in a loop to an islet enhancer. We also clarified that loops are prominently enriched for CTCF sites, in addition to active enhancer and promoter sites, and that these results are consistent with Hi-C data from other studies such as Rao et al using the same methodology.**
- **Substantially revised the main text, figures and tables to provide more clarity and consistency across the entire manuscript.**

Responses to individual comments are listed in line below.

Reviewers' comments:

Reviewer #2 (Remarks to the Author):

Greenwald et al have responded to the majority of my comments. However, some key issues remain. I agree that it is important to assign genes to enhancers and the use of HiC is important. However, the number of samples used in this study for HiC is very small and a larger number of samples is needed to make the data representative for humans, including different genotypes, and it would also be much better to include both donors with type 2 diabetes and controls. For example, only 40% of loops they present and use for analysis were identified in more than one sample. I appreciate that this represents the first high resolution 3D chromatin map of islets. However, I believe it should be done in a larger number of islet donors to merit publication in Nature Communications. In particular, since the data are used to study genetics, one would like

donors representing each genotype for the studied variants.

In their response to comment 1, they write “We agree with the reviewer that multiple studies have demonstrated the importance of islet enhancers in genetic risk of T2D. In fact, this was the primary motivating factor behind the present study - the target genes of islet enhancer activity are largely unknown, which is necessary to understand how enhancers influence T2D risk.”. However, to understand how enhancers influence T2D, they would need to perform HiC in larger number of samples including subjects with different genotypes for the T2D risk SNPs not just perform fine mapping and/or perform these studies in enough samples from donors with T2D and controls.

We appreciate the reviewer’s comment. The primary goal and novelty of this study was to prioritize causal variants and candidate target genes of T2D risk signals by combining genetic fine-mapping, islet chromatin, and expression QTL mapping – and a reference map of chromatin loops using Hi-C assays in islet samples allowed us to accomplish that goal. We also demonstrated that islet enhancers in chromatin loops to genes are correlated with islet-specific gene expression patterns, and that genetic variants in islet enhancers are significantly correlated with expression level of genes in loops relative to other genes. Our results clearly demonstrate that a reference map of chromatin loops can be used to inform how genetic variants in enhancers affect gene regulation and disease risk. Previous studies of Hi-C and related assays have generated reference maps of chromatin loops using a few representative samples to annotate target genes of enhancers and disease risk variants, but few if any have combined chromatin looping data with genetic fine-mapping, allelic imbalance and expression QTL mapping to prioritize disease genes as we have done here.

As far as we are aware, there have been no studies to date that have assayed Hi-C in large sample numbers from a single primary tissue. Our loop numbers are broadly consistent with the findings of other Hi-C studies that used the same loop calling methods, and it is unlikely that adding additional samples will drastically change the loops we identify. For example, a study including co-authors of this study (Greenwald et al, bioRxiv 2018; <https://doi.org/10.1101/352682>) generated Hi-C data from a family-based cohort of iPSC-derived cells and determined the effects of genetic variants on loop calls, which revealed that regulatory variants have at best subtle effects on chromatin loop signal. In other words, genotype by and large does not appear to affect whether a loop exists or not, and an increasing diversity of genotypes for islet chromatin variants is unlikely to have a meaningful impact on the identification of chromatin loops.

We have demonstrated that variants with significant allelic imbalance in ATAC-seq signal have concordant allelic effects on Hi-C signal. In response to the reviewer’s comments, we substantially expanded these analyses in the revised manuscript to map allelic effects on Hi-C signal for all candidate variants in enhancers at known and putative T2D loci. In total, we were able to generate Hi-C allelic imbalance statistics for the majority (60%) of all candidate T2D variants tested for ATAC-seq imbalance. This demonstrates that T2D enhancer variant genotypes are generally well represented in the set of samples assayed in this study. We included these data in Supplementary Table 6 and 7, which should be informative for interpreting the mechanisms of these variants.

Determining the effects of genetic variants on chromatin looping more broadly (e.g. through QTL mapping) would ultimately require deeply sequenced Hi-C data from ~20 samples, if not many more, given the very small effect sizes of common variants on chromatin looping. Similarly, determining differences in chromatin looping between non-diabetic and diabetic donors would require collection of numerous diabetic samples which are scarcely available. As the T2D risk variants we are studying largely affect physiology (e.g. insulin secretion measures), diabetic donors are not necessary to study their function. Performing either or both of these experiments would take years for sample collection, assay and analysis and likely cost hundreds of thousands of dollars, and therefore truly exceeds the scope and size of this study. Importantly, neither is ultimately necessary to achieve the primary goal of our study, which is to define a reference map of islet chromatin loops. We have modified the discussion to highlight that these experiments are certainly of interest to future studies:

Lines 381-384: “Future studies of chromatin looping generated across larger numbers of samples will enable a greater understanding of risk variants effects on looping directly as well as correlative relationships with gene expression and other molecular phenotypes.”

Regarding their response to comment 2: On page 6 line 170-171, I believe they added the wrong references? Should references for eQTLs be included instead?

The references are for Varshney et al PNAS (ref # 7) and van de Bunt et al Plos Genetics (ref # 20), which were the two islet eQTL datasets we used in this study.

Regarding their response to comment 3-4:, they have now added references to additional studies which have used ATAC-seq in human islets and in sorted alpha and beta cells. They also correlated their ATAC-seq data with published data and present these correlations in Figure S1B. They use merged ATAC-seq peaks for 4 samples in their analyses (n=105,734). In Table S1, they now present the number of reads and peaks per sample. There is a large variation in the number of reads (17,225,636-73,136,422 reads) and peaks (45,453-76,833 peaks) for the samples where the number of reads clearly affect the number of identified peaks per sample.

I do not believe that this approach (where peaks are merged when the technical variation is very large) is very good. This means that if a peak is present in one person it is used for analysis.

We appreciate the reviewer’s concern and agree that consideration needs to be taken in terms of the reproducibility of chromatin sites. As the reviewer notes, we have performed extensive comparisons to external data which broadly demonstrate the quality of our sites, and we also demonstrate high concordance in peak calls across our samples. This same approach of merging concordant peak calls across samples has been widely used in other studies of chromatin in islets and other tissues (e.g. Thurner et al 2018, Ackerman et al 2017, Pasquali et al 2014 to list a few). Alternate approaches - for example, by first pooling reads across samples and then calling peaks - have their own caveats such as the reduced ability to distinguish individual sites that are close together and difficulty in establishing accurate peak boundaries, in addition to a reduced ability to

identify true peaks that are present in a single or small number of samples. As the islet samples are from cadaveric donors across a range of age, sex, weight and genetic background, there are biological reasons why we would expect to find some sites that are active in few samples.

To address the reviewer's concern, we performed additional analyses to determine the reproducibility of sites in our 4 islet ATAC-seq samples to sites in published islet ATAC-seq, islet TF ChIP-seq and chromatin state data. Among sites identified in just 1 of our 4 samples, 89% were present in sites from published ATAC-seq data, and we have included this in Supplementary Figure 1C. Furthermore, sites present in 1 of 4 samples that weren't identified in published ATAC-seq data were still strongly and significantly enriched for islet enhancer sites (obs.=48%, exp.=8.8%, $P=2.2 \times 10^{-16}$) and islet TF ChIP-seq sites (obs.=7.3%, exp.=1.9%, $P=2.2 \times 10^{-16}$) (shown in Figure R1 on right) and therefore many of these singletons likely also represent functional sites.

Figure R1

We therefore feel it is appropriate to report all sites and include an additional annotation for the number of samples each site was identified in, which will enable sub-setting of chromatin sites based on different levels of reproducibility and help inform future analyses of the data. We updated Supplementary Data 1 to include the number of ATAC-seq samples that each enhancer was identified in.

Finally, in response to the reviewer comment, to then provide increased confidence in fine-mapped T2D enhancer variants we considered only variants overlapping ATAC-seq sites in >1 sample. Among the 263 enhancer variants at known T2D signals only 13 overlapped a single ATAC-seq sample, and just 12/223 enhancer variants at putative T2D signals overlapped a single ATAC-seq sample. We removed these enhancers from our analyses of T2D variants, updated Supplementary Table 6 and 7, and provided description of these changes in the main text:

Lines 231-234: *“Fine-mapping including functional priors improved causal variant resolution at these 30 signals, which on average had 3.5 candidate variants overlapping an islet enhancer and an ATAC-seq site from >1 sample (Figure 3F, Supplementary Data 1, Supplementary Table 6).”*

Lines 238-242: *“Outside of known loci, we identified an additional 127 loci genome-wide where fine-mapping identified a putative T2D risk variant that overlapped an islet enhancer and an ATAC-seq site from >1 sample (Supplementary Figure 3C, Supplementary Data 1, Supplementary Table 7; see Methods).”*

Additionally, published ATAC-seq data were used for allelic imbalance and to test if T2D SNPs are located in open chromatin regions (Table S9). In Table S9, one problem is that the majority IGF2BP2 SNPs presented do not seem to be located in open chromatin regions based on majority of ATAC-seq data and do not seem to have any allelic imbalance (except rs10428126,

which seems to give allelic imbalance but is only located in ATAC-seq peaks/open chromatin region of 7 of 23 donors)?

We appreciate the reviewer's comment and the opportunity to clarify our results. The underlying premise of genetic fine-mapping is that there is one causal variant underlying each risk signal, and the other variants are merely in linkage disequilibrium with the causal variant. Our genetic fine-mapping narrowed the set of possible causal variants at the *IGF2BP2* signal down to those listed in the table (now Supplementary Table 6), which we then further prioritized using allelic imbalance analyses. The fact that only one of the candidate variants (rs10428126) at the *IGF2BP2* signal then has evidence for allelic imbalance supports that it is likely the causal variant, and the others are probably not causal. Likewise, we find evidence for only one variant per T2D signal on average with allelic imbalance (as shown in Supplementary Table 6), and these are likely causal variants for their respective signals.

With regard to rs10428126, as the reviewer notes we identified allelic imbalance at this variant in islet ATAC-seq signal as well as allelic differences in luciferase gene reporter activity, demonstrating that this variant is functional in islets. Furthermore, this variant has been

reported as a chromatin QTL (caQTL) in an independent set of islet samples (Khetan et al 2018), supporting it is both in a *bona fide* ATAC-seq site and also has functional effects on the site, and we have added this reference in our study. The identification of this site in 7/23 donors confirms that it is reproducible across samples (i.e. not just found in 1 or 2 samples). ATAC-seq data we have generated from an additional four islet samples clearly confirms that this site is active in islets (Figure R2 on right).

Figure R2

This site is also bound by multiple islet TFs (NKX2.2, PDX1), and our luciferase reporter data showing increased activity compared to empty vector confirms that this region functions as an enhancer in beta cells. There are multiple potential biological explanations as to why this site isn't found in all/most ATAC-seq samples; for example, because the site is highly genotype-dependent (as evidenced by allelic imbalance and caQTL) it will have variable activity across samples.

Together multiple lines of evidence clearly support that this is both a functional enhancer in islets and that rs10428126 has allelic effects on this enhancer.

Regarding their response to comment 5: I do not find any definition of low methylation regions (LMRs) in the paper? What absolute percentage methylation does it represent?

We appreciate the reviewer's comment allowing us to clarify this definition. We used low methylation regions (LMRs) identified in a study by Thurner et al 2018, which they defined in their paper as "methylation ranging from 10-50% and containing fewer than 30 CpG sites". We have added this definition to the methods:

Lines 614-615: *“In this study, LMRs were defined as having methylation ranging from 10-50% and containing fewer than 30 CpG sites.”*

Regarding their response to comment 6: I appreciate that they have toned down their conclusions. The functional IGF2BP2 data are interesting. But the IGF2BP2 ATAC-seq data in Table S9, are not very supportive (see comment 3-4 above).

We appreciate the comment, and have provided a more detailed response to this comment above. In brief, our data identify a single enhancer variant at the *IGF2BP2* signal in a reproducible ATAC-seq site which has allelic imbalance in islet ATAC-seq signal and allelic effects on islet enhancer activity, and which is directionally-consistent with the *IGF2BP2* expression QTL and mouse functional data. Given the assumption that a single variant is causal for a given T2D risk signal, we feel this is fully supportive that we have identified the likely causal variant and risk mechanism underlying this signal in reducing islet enhancer activity and *IGF2BP2* activity.

Regarding their response to comment 7: Could you please include number of aligned reads and number ATAC-seq peaks generated for frozen and fresh samples for ISL3, as well overlapping peaks as not just a correlation.

We have provided the information regarding ISL3 data from frozen and fresh islets in Supplementary Table 1.

Regarding their response to comment 15: I am not convinced about the method/analysis used here, where they correlated the number of loops annotated to a gene using their HiC data set of 3 samples with average level of expression in 118 islets samples in a different data set. Gene expression of their own samples should be used here and individual samples should be used for correlation analysis between HiC and expression data in a bigger number of samples.

We thank the reviewer for the comment. Exploring the correlative relationship between chromatin interactions and gene expression in this manner would require Hi-C data from many additional samples, which as described in our response to comment #1 is both not in line with the goals of our study and outside the scope of what is practically possible. Our goal was to define a reference map of chromatin loops in islets in order to annotate candidate target genes of enhancer elements. In order to then relate this map to gene expression, we used a large reference set of expression data from independent islet samples. The results of these analyses clearly demonstrate a relationship between enhancer looping and gene expression level in islets, and furthermore that this relationship is highly tissue-specific when compared to gene expression in other, non-islet GTEx tissues. We have modified the discussion to highlight that these additional analyses using data generated across many matched samples are of interest to continued studies:

Lines 381-384: *“Future studies of chromatin looping generated across larger numbers of samples will enable a greater understanding of risk variants effects on looping directly as well as correlative relationships with gene expression and other molecular phenotypes.”*

Reviewer #3 (Remarks to the Author):

In this study Hi-C data is generated from human pancreatic islets, and this data is used to better determine the promoters distal enhancers actually associate with and thus which target genes variant containing enhancers might actually be regulating. This is important and valuable insight. By combining this data with eQTL data they identify 8 enhancers that interacted with genes with T2D eQTLs, including CAMK1D, ABCB9, C2CD4B, and IGF2BP2.

They then focus on IGF2BP2, and knock-out this gene in mouse beta-cells using the RIP2-Cre driver. They find IGF2BP2 (Imp2 in mice) deficient mice have HFD diet dependent glucose intolerance.

This revised version of the manuscript address my previous concerns and seems to largely address the concerns of the other previous reviewer. I know think it is suitable for publication.

We thank the reviewer for their positive comments, and especially want to highlight their assertion that we have also addressed the comments of Reviewer #2.

Reviewer #4 (Remarks to the Author):

Comments on Greenwald et al: Pancreatic islet chromatin accessibility and conformation defines distal enhancer networks of type 2 diabetes risk (180095 Nature Communications)

Comments for the authors:

The authors perform ATAC-seq and Hi-C, and in combination with published ChIP-seq, GWAS and eQTL data use this to map putative regulatory elements in pancreatic islet cells, and to link these regulatory regions to potential target genes. The authors thereby generate a catalogue of regulatory elements and their putative target genes in pancreatic islets, which they characterize using chromatin state analyses and transcriptome data. Target genes were found to be enriched in genes involved in protein transport and secretion pathways. The effect of two risk allele variants on transcription is assayed in reporter gene assays. Finally, the authors delete one target gene of a type 2 diabetes (T2D) risk allele, the Imp2 gene (IGF2BP2 in human) in mice, and show that the absence of this gene results in defects in glucose-stimulated insulin secretion.

The experimental approach of linking disease-associated sequence variants to target genes using chromosome conformation capture techniques is timely, and highly relevant to uncover regulatory mechanisms that underlie disease, as in this case T2D. This is especially true in the light of the discovery that the vast majority of GWAS hits map to regions with regulatory potential located in the non-coding genome (Maurano et al., 2012). The experiments in this study are conducted and controlled adequately. However, unfortunately the manuscript contains a series of mistakes which add up to the impression of a hastily put together and as a consequence, very difficult to read manuscript. Some of the findings presented merely confirm previous results (for example the finding that gene expression levels correlate positively with the number of interacting enhancers, which has been shown before in several papers using

Promoter Capture Hi-C). I also have concerns about the very high number of chromatin loops that involve promoters and enhancers; this is in disagreement with previously published studies (see below). In conclusion, the manuscript in its current form is not suitable for publication, and I am not convinced that the novelty of the findings presented here justifies publication in Nature Communications. I would therefore advise against publication.

We thank the reviewer for their comments on our manuscript. The primary goal and novelty of this study was to prioritize causal variants and candidate target genes of T2D risk signals by integrating genetic fine-mapping, islet chromatin, allelic imbalance mapping and expression QTL data. There have been few studies which have combined this breadth of approaches in order to predict the causal variants and genes involved in complex disease risk, and no published studies we are aware of thus far in the context of T2D. Furthermore, for one of these candidate genes we demonstrated that conditional inactivation in a mouse model produces a diabetic phenotype that is directionally consistent with the human genetic data, which both confirms our approach and provides a novel gene and mechanism involved in T2D pathogenesis. While findings such as the correlation between enhancer loops and gene expression level may be similar to other reports in different cells and tissues, they have first not been demonstrated in islets and second serve to justify our use of enhancer loops to prioritize genes of T2D risk signals.

In response to the reviewer comments we have:

- **Performed extensive revisions to the main text, figures and tables to improve clarity of the manuscript and that fully address the changes suggested by the reviewer.**
- **Provided clarification of enhancer and promoter looping and made extensive revisions to the analyses, text and figures describing these results.**

These revisions are detailed in response to each comment below.

Major points:

1.) Introduction, lines 73-75: "...we generated the first high-resolution, genome-wide map of 3D chromatin architecture in pancreatic islets, and used this map to annotate islet enhancers defined using ATAC-seq assays and published ChIP-seq data."

This is not correct. The 3D chromatin architecture map is not used to annotate enhancers - this is done using ATAC-seq and ChIP-seq. The 3D chromatin architecture map is used to link these annotated enhancers to their putative target genes.

We apologize for the confusion caused by the choice of wording. We meant to say that we were using the 3D chromatin architecture map to annotate the target genes of islet enhancers (which we defined using ATAC-seq/ChIP-seq). We have clarified this in the text:

Lines 38-41: "In this study we generate a high-resolution map of islet chromatin loops using Hi-C assays in three islet samples and use loops to annotate target genes of islet enhancers defined using ATAC-seq and published ChIP-seq data"

2.) In the methods (line 443), the CMM states are explained as "...Active Enhancer 1 (EnhA-1; H3K27ac), Active Enhancer 2 (EnhA-2; H3K27ac, H3K4me1)". This does not match with what is shown in Figure S1C. The enrichment for H3K4me1 is stronger for EnhA-1 than for EnhA-2, both in absolute terms, and in comparison to H3K27ac within the respective enhancer categories.

We apologize for the confusion, and appreciate the reviewer catching this error – we mistakenly swapped the labels of EnhA1 and EnhA2 in the methods. EnhA1 has both strong H3K27ac and H3K4me1 signal, whereas EnhA2 has strong H3K27ac signal only. We have updated the methods:

Lines 490-498: "We defined chromatin states from ChIP-seq data using ChromHMM¹⁶ with a 9-state model, as calling larger state numbers did not empirically appear to identify additional states. We assigned the resulting states names based on patterns previously described in the NIH Roadmap and ENCODE projects – CTCF (CTCF), Transcribed (Txn; H3K36me3), Active promoter (TssA; H3K4me3, H3K4me1), Flanking promoter (TssFlnk; H3K4me3, H3K4me1, H3K27ac), Weak/Poised Enhancer (EnhWk; H3K4me1), Active Enhancer 1 (EnhA1; H3K27ac, H3K4me1), Active Enhancer 2 (EnhA2; H3K27ac), and two Quiescent states with low signal which we merged together (Quies/low; low signal for all assays)."

3.) Figure S1C: is the chromatin state 'Quiesc' duplicated here on the y axis? This state seems to be devoid of any chromatin on the x axis. Why is it called 'quiescent', which is a rather unusual description for a chromatin state (quiescence in its usual definition refers to a state in which cells do not divide but retain the ability to re-enter cell proliferation). Why not call this state 'background' or 'no mark'?

We used the designation of 'Quies' for regions without ChIP-seq signal as this is the terminology the Roadmap Epigenomics and ENCODE consortia used to define states with low/no signal. While we agree with the reviewer that this may not be the most accurate term to describe an absence of signal, we simply used it in order to provide consistency with what has been reported in previous studies. We have updated the name of this state to 'Quies/low' throughout the paper in both the text and figures in order to both facilitate cross-referencing state names with Roadmap and other studies, as well as to clarify that this state is defined by having low signal.

As shown in Supplementary Figure 1D we observed two states with low signal, so we assigned them both the 'Quies/low' name and merged their states. We clarify this in the methods:

Lines 490-498: "We defined chromatin states from ChIP-seq data using ChromHMM¹⁶ with a 9-state model, as calling larger state numbers did not empirically appear to identify additional states. We assigned the resulting states names based on patterns previously described in the NIH Roadmap and ENCODE projects – CTCF (CTCF), Transcribed (Txn; H3K36me3), Active promoter (TssA; H3K4me3, H3K4me1), Flanking promoter (TssFlnk; H3K4me3, H3K4me1, H3K27ac), Weak/Poised Enhancer (EnhWk; H3K4me1), Active Enhancer 1 (EnhA1; H3K27ac, H3K4me1), Active Enhancer 2 (EnhA2; H3K27ac), and two Quiescent states with low signal which we merged together (Quies/low; low signal for all assays)."

4.) Are figures S1D, S1E and S1F mentioned in the text? If not, why are they there?

We apologize for the confusion. We had previously referenced these results as Supplementary Figure 1 on line 105, and have now separated out the description of each specific figure panel next to their respective results in the main text on line 102-105:

Lines 99-104: *“We identified 44,860 active enhancers which, in line with previous reports^{4,17}, were distal to promoters (Supplementary Figure 1E), more tissue-specific (Supplementary Figure 1F), overlapped islet transcription factor ChIP-seq sites (Supplementary Figure 1G), and preferentially harbored sequence motifs for FOXA, RFX, NEUROD and other islet transcription factors (Supplementary Table 2).”*

5.) Lines 141-142: "Conversely, the promoter regions of 8,448 genes had at least one loop to an enhancer element."

This seems extremely high. How many loops are there that do not involve a promoter or enhancer, out of the 11,924 loops identified in total? Most loops in deep-sequenced Hi-C data appear to be between CTCF sites (for example Rao et al., Cell 2014), and only a minority involve promoters or enhancers. How do the authors explain this discrepancy to published data?

We appreciate the reviewer's comment and highlighting this issue. We used the same loop calling method and approach as in Rao et al., and when considering elements that overlap a loop directly CTCF sites are by far the most enriched state in loop anchors (see Figure 1D), and CTCF-CTCF is the third most frequently interacting pair (N loops=2,840) after CTCF-EnhA1 (N loops=3,226) and EnhA1-TssA (N loops=3,086). We note that in the Rao et al. paper they do find many enhancer and promoter interactions, and indeed mention directly in the abstract that "loops frequently link enhancers and promoters". Our data are therefore consistent with the findings of other studies. We have added a new supplemental figure panel (Supplementary Figure 2C) which shows the number of loops containing pairs of sites, in addition to the enrichment of pairs shown in Figure 1E.

Our approach to defining candidate enhancer and gene promoter relationships originally used a flanking window (25kb) around chromatin loop boundaries, thus resulting in a larger set of candidate enhancer-promoter interactions. We extensively revised the relevant sections of the manuscript to instead describe enhancer and promoter links based on direct overlap with chromatin loops. In these results there are 3,022 enhancers in a direct loop to a gene promoter, and 2,028 promoters in a direct loop to an enhancer. We also substantially revised the analyses relating active enhancer and promoter loops to gene expression and expression QTLs to use sites directly overlapping a loop, and moved the previous analyses of sites within 25kb of loop boundaries to Supplementary Figure 2. We note that even when using sites within 25kb of a loop we find strong and significant correlation between enhancer looping and gene expression and expression QTLs.

Lines 125-130: *“Nearly half (48.7%) of all islet accessible chromatin sites were within 25kb of an anchor, and 16.8% directly overlapped an anchor. Sites most enriched (empirical $P < 1.5 \times 10^{-4}$) for direct overlap with chromatin loop anchors were those in CTCF binding (7.5-fold) states, followed by active promoter (TssA: 3.9-fold; TssFlnk: 3.3-fold), and active enhancer (EnhA1: 2.4-fold) states (Figure 1D).”*

Lines 141-196: “We next used chromatin loops to annotate candidate relationships between distal islet enhancers and their potential target genes genome-wide (see Methods). We identified 6,278 islet active enhancers that mapped directly in a chromatin loop anchor and, of these, 3,022 enhancers were in a loop to a gene promoter (**Supplementary Figure 2D, Supplementary Table 3**). Conversely, the promoter regions of 2,028 genes had at least one direct loop to an active enhancer element (**Supplementary Figure 2E, Supplementary Table 4**). Of these 2,028 genes, 952 (47%) had chromatin loops to multiple active enhancers (**Supplementary Figure 2E**). Genes directly looped to multiple enhancers were enriched for processes related to transcription factor activity and gene regulation, signaling and stimulus response, protein transport and insulin signaling (**Supplementary Table 5**), and also included genes critical for islet function such as *ISL1*, *FOXA2*, *NKX6.1*, and *MAFB* (**Supplementary Table 4**). At many loci enhancers looped to gene promoters over long distances; the average distance between interacting enhancer and gene promoter pairs was 165kb, with 13.9% (532) over 500 kb and 3.6% (138) over 1 Mb (**Figure 2A**). For example, there were four chromatin loops at the *MAFB* locus, including two direct loops between enhancers and the *MAFB* promoter region over 1 Mb distal (**Figure 2B**). These results define candidate target genes for thousands of distal enhancer elements in islets.

We examined the relationship between active enhancer looping and target gene expression. We compared our map of islet enhancer candidate target genes defined from islet chromatin loops to gene expression levels in independent RNA-seq data from pancreatic islet samples²⁰ and 53 tissues in GTEx release v7 data²¹. A significantly higher proportion of genes expressed in islets had at least one enhancer loop compared to non-islet expressed genes ($\ln(\text{TPM}) > 1$; $\text{expr} = .13$, $\text{non-expr} = .05$, $\chi^2 P < 2.2 \times 10^{-16}$). Genes with increasing numbers of enhancer loops had, on average, higher expression level in islets (Spearman $r = .13$, $P < 2.2 \times 10^{-16}$), with the highest expression among genes with 6 or more loops (median = 19.1 TPM) (**Figure 2C**). We measured the relative expression level of genes in islets and 53 GTEx tissues normalized across tissues (see Methods), and again observed a significant relationship between enhancer loops and relative islet expression level (Spearman $r = .084$, $P < 2.2 \times 10^{-16}$) (**Figure 2D**). In addition, the number of islet enhancer interactions was a significant predictor of higher relative gene expression level in islets ($r = .14$, $P < 2.2 \times 10^{-16}$) but not of relative expression level in the 53 other tissues (**Figure 2E**). We observed similar correlations between distal enhancers and islet gene expression when considering sites within a 25kb region around each loop anchor, suggesting that these relationships extend beyond anchor boundaries (**Supplementary Figure 2F, 2G**). These results suggest that distal islet enhancer chromatin loops are correlated with islet-specific gene expression patterns.

We next determined the effects of genetic variants in islet enhancers on target gene regulation. We generated expression quantitative trait locus (eQTL) data from 230 islet RNA-seq samples by combining summary statistics from two published studies through meta-analysis^{7,20} (see Methods). We identified variants overlapping classes of islet regulatory elements genome-wide. We then quantified the eQTL association of these variants to target genes determined by their proximity to nearby genes and from chromatin loops (see Methods). As expected, we observed the strongest eQTL evidence for active promoter and enhancer variants proximal to genes (TssA: median $-\log_{10}(P) = .64$; EnhA proximal: median $-\log_{10}(P) = .50$) (**Figure 2F**). For variants in distal enhancers, we observed significantly stronger evidence for islet eQTL association with genes in direct loops to the enhancer relative to non-loop genes (EnhA loop median = .35, EnhA non-loop median = .32, Wilcox $P = 8.2 \times 10^{-5}$), even when matching based on gene distance to the enhancer (EnhA non-loop matched, Wilcox $P = .022$) (**Figure 2F**). We observed similar eQTL enrichment among enhancer variants looped to gene promoters when considering sites within 25kb of a loop anchor (**Supplementary Figure 2H**). These results suggest that genetic variants in distal islet

enhancer elements are preferentially correlated with the expression level of genes in chromatin loops.”

To identify candidate target genes of T2D enhancer variants we used the 25kb window definition. The rationale behind this definition was to capture a larger initial set of candidate genes which we then prioritized further through expression QTL mapping, and which we further justify based on the significant correlation between enhancers and gene expression and expression QTLs when considering sites within 25kb of a loop. We clarified the description of these results in the main text and methods, included additional text and supplemental figure panels for several candidate gene promoters in direct loops to T2D enhancer variants (Supplementary Figure 4), and also revised Supplementary Table 8 to include annotation of which candidate genes are in direct loops with T2D enhancer variants:

Lines 268-274: “In order to identify genes affected by T2D risk variants in enhancers, we used a tiered strategy whereby we first identified candidate target genes of these enhancers using chromatin looping and promoter-proximity, and then further prioritized candidate genes cis-regulated by T2D enhancer variants using eQTL mapping. For each T2D enhancer signal (from **Figure 3D**), we identified candidate genes based on whether an enhancer variant was within 25kb of either a chromatin loop to the gene promoter or the gene promoter itself (see Methods).”

Lines 278-286: “At several loci, loops implicated candidate target genes highly distal (>500kb) to T2D enhancer variants. For example, at the 3q27 locus T2D variants directly looped to the *TPRG1* promoter 900kb distal (**Supplementary Figure 4A**), and at the 10p13 locus T2D variants looped to the *OPTN* and *CCDC3* promoters 840 kb distal (**Supplementary Figure 4B**). In additional examples, T2D enhancer variants at the 11p15 locus near *KCNQ1* looped to the *CDKN1C* promoter as well as to the *INS/IGF2* locus 700kb distal (**Supplementary Figure 4C**), and T2D enhancer variants at the 10q22 locus near *ZMIZ1* looped to the *POLR3A* locus 1MB distal (**Supplementary Figure 4D**).”

6.) Figure 2E is confusing. First, the proximal and Enh Distal (target) categories are referred to as dark blue and blue, respectively - these look the same shade of blue to me. Second, two of the categories on the x axis are called the same 'Enh Distal (no target)'. My understanding is that both describe genes not involved in loops, but one is distance-matched (the one referred to as light blue; maybe 'turquoise' is better?) - this needs to be pointed out in the figure itself.

We apologize for the confusion. We have clarified the main text, figure and legend (now in Figure 2F) in terms of both the color scheme used as well as the description of each category of variants:

Lines 186-192: “As expected, we observed the strongest eQTL evidence for active promoter and enhancer variants proximal to genes (*TssA*: median $-\log_{10}(P)=.64$; *EnhA* proximal: median $-\log_{10}(P)=.50$) (**Figure 2F**). For variants in distal enhancers, we observed significantly stronger evidence for islet eQTL association with genes in direct loops to the enhancer relative to non-loop genes (*EnhA* loop median=.35, *EnhA* non-loop median=.32, Wilcox $P=8.2 \times 10^{-5}$), even when matching based on gene distance to the enhancer (*EnhA* non-loop matched, Wilcox $P=.022$) (**Figure 2F**).”

Lines 1071-1076: “(F) Gene expression QTL p-values for genetic variants in gene promoters (TssA; red), enhancers proximal to gene promoters (Enh. proximal; pink), enhancers in chromatin loops to the gene promoter (Enh. loop; dark blue), and enhancers not in chromatin loops to the gene promoter for both all enhancers (Enh. no-loop; light blue) and enhancers distance-matched with looped genes (Enh. no-loop matched; grey).”

7.) Lines 293-295: "We further validated that the risk allele at rs10428126 reduced islet enhancer activity using gene reporter assays..."
In this case (Fig. S5A), as in Figure 3H, the Ref alleles drive higher reporter gene expression. However, here, the authors conclude that "that the risk allele at rs10428126 reduced islet enhancer activity", whereas "...the risk allele at rs7732130 increased enhancer activity..." (line 241). Are both these statements correct?
Better labelling of the figures would help to avoid confusion. Is the Ref allele the respective risk allele in both Figs 3H and S5A?

We apologize for the confusion. The risk allele can either be the reference or alternate allele depending on the variant – in other words, for some T2D variants the risk allele is the reference allele, and for some the risk allele is the alternate allele. In the case of rs10428126, the risk allele is the alternate allele, which has reduced enhancer activity in Supplementary Figure 5B. In the case of rs7732130, the risk allele is the reference allele, which has higher enhancer activity in Figure 3H. We have clarified both in the main text and the figure legends for both Figure 3 and Supplementary Figure 5 which allele (ref or alt) is the risk allele for a given variant:

Lines 258-260: “Among the 19 novel imbalanced variants, rs7732130 at 5p13 is causal for T2D (PPA=98%) and the T2D risk (and reference) allele G increased chromatin accessibility ($P=7.1 \times 10^{-4}$).”

Lines 332-334: “We observed significant evidence ($FDR < .1$) for allelic imbalance at rs10428126 (binomial $P = .001$) where the T2D risk (and alternate) allele C had reduced accessibility”

Lines 1096-1098: “(H) rs7732130 has allelic effects on enhancer activity in the islet cell line MIN6 (N=3), where the T2D risk allele and reference (ref) G has higher activity than the alternate (alt) allele A.”

Supplementary Figure 5 legend: “(B) T2D variant rs10428126 has allelic effects on islet enhancer activity in the islet cell line MIN6 where the T2D risk and alternate (alt) allele C has reduced activity than the reference (ref) allele ($P = .001$; $N = 3$).”

8.) The Imp2 inactivation experiment addresses the role of IMP2/IGF2BP2 in glucose metabolism, but I cannot see how this is directly linked to an enhancer risk variant? The gene deletion will no doubt result in more dramatic downregulation of gene activity than a sequence variant in the enhancer?

The observation that T2D risk variants at this locus reduce islet enhancer activity and IGF2BP2 expression led us to hypothesize that reduced IGF2BP2 in islets contributes to a diabetic phenotype. We tested this hypothesis using conditional inactivation of IGF2BP2 activity in mouse beta cells, which demonstrated that absence of IGF2BP2 activity impaired insulin secretion. While the reviewer is correct that a gene knockout

will produce a more profound effect than an enhancer variant, these results nonetheless provide a clear validation of our hypothesis. Given the small effect of disease risk variants it isn't clear that manipulating the specific T2D enhancer variant would produce a measurable phenotype either in a cell model or an animal model. We have revised the manuscript to better explain the rationale for the *IGF2BP2* loss-of-function experiment:

Lines 344-347: "As T2D risk alleles at the *IGF2BP2* locus are correlated with reduced islet chromatin accessibility, enhancer activity and *IGF2BP2* expression as well as reduced insulin secretion phenotypes²⁴, we hypothesized that reduced activity of *IGFBP2* would contribute to a diabetic phenotype in islets."

9.) What is shown in Figure 5D and E? According to the figure legends for 5D and 5E, pretty much the same thing?

Lines 879-881: "(D) 1 g/kg glucose was administered intraperitoneally after overnight fasting of 12-week-old *Imp2ff* (black; N=10) and *imp2ff/RIP2-Cre* (red; N=10) male mice maintained on normal chow diet (NCD)."

Lines 882-884: "(E) 1 g/kg glucose was administered intraperitoneally after overnight fasting to 12-week-old *Imp2ff* (black; N=9) and *Imp2ff/RIP2-Cre* (red; N=9) male mice maintained on NCD."

By contrast, lines 315-318 in the main text state: "When challenged with an intraperitoneal glucose injection, HFD-fed, but not NCD-fed, *Imp2ff/RIP2-Cre* mice exhibited significantly higher glucose and lower insulin levels than *Imp2ff* mice (Figure 5D,E).

I suspect (D) shows data from NCD-fed mice, and (E) shows data from HFD-fed mice?

We apologize for the confusion created by not correctly labelling the feeding type in Figures 5D and E; the reviewer is correct that Figure 5D shows data from NCD-fed and Figure 5E HFD-fed mice, and we have updated the figure legend accordingly:

Lines 1125-1127: "(E) 1 g/kg glucose was administered intraperitoneally after overnight fasting to 12-week-old *Imp2ff* (black; N=9) and *Imp2ff/RIP2-Cre* (red; N=9) male mice maintained on HFD."

Minor points:

10.) Affiliation: 13. Present affiliation: Center for Epigenomics, UC San Diego, La Jolla CA - there is no 13 associated with any of the authors from what I can see. Also, is this not identical to 7 (Center for Epigenomics, UC San Diego, La Jolla CA)?

Affiliation 13 was included for author Allen Wang, who has taken up a new position at the UCSD Center for Epigenomics since the submission of the manuscript. For clarity, we have now simply used affiliation 7 to denote this.

11.) Abstract, lines 43-45: "We identified target genes for thousands of distal islet enhancers, many interacting over 1Mb distances, and genes interacting with enhancers were correlated with islet-specific expression patterns."

An example where more precise language would help, see also points 12 and 13 below. Should this be: "We identified target genes for thousands of distal islet enhancers, many interacting

over 1Mb distances, and found that genes interacting with enhancers were correlated with islet-specific expression patterns."

We appreciate the clarification and have implemented it in the abstract. Note that due to journal requirements we have also shortened the abstract length and changed to present tense.

Lines 41-43: "We identify candidate target genes for thousands of islet enhancers, many interacting over 1Mb, and find that enhancer looping is correlated with islet-specific gene expression."

12.) Abstract, lines 48-50: "We defined target genes of these T2D islet enhancer signals using chromatin looping and islet eQTL mapping, and target genes were specifically enriched in protein transport and secretion pathways."

Should this be: "We defined target genes of these T2D islet enhancer signals using chromatin looping and islet eQTL mapping, and found that target genes were specifically enriched in protein transport and secretion pathways."

We appreciate the clarification and have implemented it in the abstract:

Lines 43-45: "We fine-map T2D risk variants affecting islet enhancers, and find that candidate target genes of these variants defined using chromatin looping and eQTL mapping are enriched in protein transport and secretion pathways."

13.) Introduction, lines 62-63: "The genes regulated by islet enhancers are largely unknown, however, impeding the discovery of disease-relevant gene networks and pathways perturbed by risk variants and novel therapeutic pathways."

As it stands this reads as "...novel pathways perturbed by risk variants."

Better would be, for example: "The genes regulated by islet enhancers are largely unknown, however, impeding the discovery of disease-relevant gene networks and pathways perturbed by risk variants and the development of novel therapeutic pathways."

We agree that this sentence could be rephrased to add clarity and have updated it to the reviewer's suggestion:

Lines 54-56: "The genes regulated by islet enhancers are largely unknown, however, impeding discovery of disease-relevant gene networks and pathways perturbed by risk variants and the development of novel therapeutic avenues."

14.) Figure S1B: ISL1_frozen, ISL2_frozen, ISL3_frozen, ISL4_frozen: do these correspond to ISL1 to ISL_4 in Figure S1A? There is an additional sample ISL3_fresh in Figure S1B?

We apologize for the confusion. Samples ISL1-ISL4 in Supplementary Figure 1A indeed correspond to ISL1-4_frozen in Supplementary Figure 1B, as these data were generated from frozen islets. ISL3_fresh is an additional ATAC-seq assay generated from fresh cells from the ISL3 sample which was included in Supplementary Figure 1B to demonstrate the high concordance between ATAC-seq data from frozen and fresh islets.

For consistency, we renamed the samples in Supplementary Figure 1B to match up with the names in Supplementary Figure 1A, and clarified 'ISL3_fresh' in the figure legend:

Supplementary Figure 1 legend: “(A) Heatmap of the Spearman correlation between ATAC-seq read coverage in merged peaks across four islet samples (ISL1-4). (B) Jaccard overlap between peak calls for the four islet samples (ISL1-4), one sample with additional data generated from fresh cells (ISL3_fresh), 19 islet samples from two published studies, and sorted alpha, beta and acinar cells from a published study.”

15.) Results, lines 94 to 96: "..., as well as strong concordance in peak calls from published data of ATAC-seq from 19 islet samples and FACS-sorted beta and alpha cells."
Is such a general statement justified here? The concordance appears to stretch over a wide range of values, and I don't see strong concordance in every example. For example, the concordance between ISL3_frozen/ISL_fresh3/ISL4_frozen with Acinar_2/Beta2/HP1443Hg19 is rather weak.

We agree with the reviewer that there is a range of concordance values and a subset of samples (Acinar_2, Beta_2, HP1443Hg19) show lower concordance with the majority of other samples. We have updated the main text to reflect this:

Lines 89-92: “We observed strong correlation in both accessible chromatin signal and peak calls across samples (**Supplementary Figure 1A**), as well as concordance with peak calls from the majority of published ATAC-seq data from 19 islet samples and FACS-sorted beta and alpha cells^{7,13,14} (**Supplementary Figure 1B, 1C**).”

16.) Figure 1A: can the authors explain why accessible chromatin is much more enriched in the EnhA1 state compared to EnhA2? The authors statement that "Accessible chromatin predominantly mapped with active enhancer and promoter states" (line 99) is true for EnhA1 and TssA, but not for EnhA2, which is less enriched than the Quies state for ATAC-seq signals.

We apologize for the confusion. This is in part because there are 44,860 EnhA1 sites which cover 17.9 Mb of sequence and 16,779 EnhA2 sites that cover 4.2 Mb of sequence, and therefore as EnhA1 covers more of the genome these sites will naturally overlap with more accessible chromatin. In addition, EnhA1 sites have both H3K4me1 and H3K27ac marks (whereas EnhA2 sites have just H3K27ac), and are more enriched for overlap with islet transcription factor ChIP-seq sites (Supplementary Figure 1G), sequence motifs for islet TFs (Supplementary Table 2), and chromatin loops (Figure 1E), suggesting they are generally more functionally active sites than EnhA2. We have updated the text to clarify that EnhA1 and TssA are the specific states enriched for accessible chromatin:

Lines 95-97: “Accessible chromatin predominantly mapped within active enhancer (EnhA1) and promoter (TssA) states (**Figure 1A**).”

17.) Lines 104 - 105: "...preferentially harboured motifs for FOXA, RFX and NEUROD and other islet transcription factors (Figure S1, Table S3).

Two issues here:

I assume this refers to Figure S1F? And why do the authors not show the data for RFX and

NEUROD in this supplemental figure, if those are indeed the most enriched motifs? In Figure S1F, the enrichment for NKX6.1, FOXA2, MAFB, PDX1 and NKX2.2 are shown.

We apologize for the confusion. There are two separate analyses being referred to in the text. The first is in Supplementary Figure 1G where we observe enrichment of active enhancer elements (EnhA1) for transcription factor ChIP-seq sites for NKX6.1, FOXA2, MAFB, PDX1 and NKX2.2. The second is in Supplementary Table 2 where we observe enrichment of sequence motifs in active enhancer elements for RFX, NEUROD, and FOXA among other factors. We have clarified these two distinct analyses and added references to the correct Figure/Table in the main text:

Lines 99-104: *“We identified 44,860 active enhancers which, in line with previous reports^{4,17}, were distal to promoters (Supplementary Figure 1E), more tissue-specific (Supplementary Figure 1F), overlapped islet transcription factor ChIP-seq sites (Supplementary Figure 1G), and preferentially harbored sequence motifs for FOXA, RFX, NEUROD and other islet transcription factors (Supplementary Table 2).”*

18.) Figure legend S1E (lines 896-897): "islet regulatory elements" - these should be called chromatin states. I can't see any justification for calling the 'Quies' state a regulatory element.

We used the term ‘regulatory elements’ to refer to accessible chromatin sites that we then annotated with chromatin states; for example, ‘Quies’ is an islet accessible chromatin site that has a Quiescent chromatin state. We appreciate the confusion in this terminology, however, and have modified the description of these data to reflect that they are accessible chromatin sites annotated with chromatin states:

Supplementary Figure 1 legend: *“(F) Percentage of ENCODE cell-types in DHS sites overlapping islet accessible chromatin sites in each chromatin state.”*

19.) Figure legend S1F (line 898): "islet regulatory elements" - these should be called chromatin states. I can't see any justification for calling the 'Quies' state a regulatory element, see point 18 above.

See response to comment 18 above, we updated the text as follows:

Supplementary Figure 1 legend: *“(E) Heatmap showing percentage of islet accessible chromatin sites in each chromatin state mapping in 200bp bins around GENCODE transcriptional start sites.”*

20.) Figure S1F: the Jaccard metric is missing (whereas it is present in for example S1B).

We have added the label as suggested to this figure panel (now Supplementary Figure 1G).

21.) Lines 125-126: "Nearly half of all islet regulatory elements were proximal to an anchor..." See points 18 and 19 above. I don't think 'regulatory elements' is correct here. The authors refer to chromatin states, which include one category (quiescent) which is devoid of any of the

assayed chromatin marks and thus cannot be assigned regulatory element status by any criteria that I can think of.

See response to comment 18 above, and we updated the text as follows:

Lines 123-137: “We next determined the relationship between islet accessible chromatin and chromatin looping. Islet accessible chromatin signal was largely localized to islet loop anchors, with the strongest signal at anchor midpoints (**Figure 1C**). Nearly half (48.7%) of all islet accessible chromatin sites were within 25kb of an anchor, and 16.8% directly overlapped an anchor. Sites most enriched (empirical $P < 1.5 \times 10^{-4}$) for direct overlap with chromatin loop anchors were those in CTCF binding (7.5-fold) states, followed by active promoter (TssA: 3.9-fold; TssFlnk: 3.3-fold), and active enhancer (EnhA1: 2.4-fold) states (**Figure 1D**). We further mapped the relationship between pairs of islet accessible chromatin sites directly connected by loop anchors (**Supplementary Figure 2C**). The most significantly enriched anchor interactions were between active enhancer and promoter states (EnhA1-TssA OR=1.28, $P=1.53 \times 10^{-37}$; EnhA1-EnhA1 OR=1.37, $P=1.87 \times 10^{-38}$; TssA-TssA OR= 1.42, $P=6.15 \times 10^{-36}$). We also observed strong enrichment for interactions between CTCF binding states (CTCF-CTCF OR=1.16; $P=1.1 \times 10^{-17}$) (**Figure 1E**). These results demonstrate that islet chromatin loops are prominently enriched for CTCF binding as well as active promoter and enhancer regions.”

22.) Line 147-148: "..., there were four distinct loops between active enhancers and the MAFB promoter, including several loops to enhancers over 1 Mb distal."
Several in this case means two. Just say two.

We have clarified that two enhancers loop to the MAFB promoter:

Lines 155-157: “For example, there were four chromatin loops at the MAFB locus, including two direct loops between enhancers and the MAFB promoter region over 1 Mb distal (**Figure 2B**).”

23.) Lines 828-829: "Multiple islet enhancers formed chromatin loops with the MAFB promoter region including several over 1MB."

Related to point 22 above. It's four enhancers, and two interactions over 1Mb. Just state it as it is: "Four islet enhancers formed chromatin loops with the MAFB promoter region including two over 1MB."

We have clarified the text to add the numbers of enhancers and loops.

Lines 155-157: “For example, there were four chromatin loops at the MAFB locus, including two direct loops between enhancers and the MAFB promoter region over 1 Mb distal (**Figure 2B**).”

24.) Figure 2B: which chromosome is shown? What is the colour code for the chromatin states? As for the chromatin state colours, are they the same as in Figure 1D? If yes, how will the readers be able to distinguish between EnhA1 and EnhA2, which seem to be the same shade of dark blue?

We apologize for not including a color legend and appreciate the reviewer highlighting this oversight, as the colors for some of the states did not match up with Figure 1 (e.g. CTCF was orange in Fig 1, green in Fig 2). We also changed EnhA2 to purple in order to help distinguish EnhA1 and EnhA2. We updated the coloring of chromatin states in the plot in order to provide consistency throughout the manuscript, and included a legend in the figure as well as the chromosome number.

25.) Figure 2B right panel: although difficult to assess at this resolution, at least significant loops appear very close to the diagonal, whereas other clearly visible interactions in this heatmap have not been called as significant. Are the authors confident in their loop-calling, especially in the light that the enrichment of promoters and enhancers at their chromatin loop anchors differs markedly from other reports (Rao et al., 2014), see point 5 above?

We used the same chromatin loop calling method and approach as in Rao et. al (HICCUPS), and therefore are generally confident that the loops represent high quality calls with respect to the current standard. Compared to other methods HICCUPS likely results in more conservative, yet higher confidence, loop calls. While some true loops are surely not called with HICCUPS, this is an unavoidable aspect of loop calling with any of the current methods as recent manuscripts have highlighted (e.g. Forcato et al Nat Meth 2017). With respect to the enrichment of loops for enhancer and promoters we have addressed this in detail in response to comment 5.

26.) Lines 116 - 118: "We merged the resulting four sets of loop calls where both anchors overlapped at 20 kb resolution." Do the authors think this kind of resolution is sufficient to analyse individual enhancer-promoter interactions, for example. A competing study in bioarchive (Miguel-Escalada et al.) uses Promoter Capture Hi-C which offers single restriction fragment resolution...

The approach used for merging anchors within 20kb resolution is the recommended application of HICCUPS, the algorithm created by Rao et. al. in their seminal paper. The enrichment of genes with islet enhancer loops for islet-specific gene expression as well as the enrichment of genetic variants within enhancers for expression QTLs to genes in loops both support that at this resolution our data identify enhancer and promoter relationships. We have been careful throughout the revised manuscript to describe these as 'candidate' relationships that would need to be validated in order to confirm they are true interactions; this would of course still be true of higher resolution data such as pChIP-C.

Furthermore, our strategy for identifying T2D-relevant relationships between enhancers and promoters consisted of first using chromatin looping data and promoter-proximity to identify a set of candidate genes and then using eQTL mapping to identify candidate genes *cis*-regulated by T2D risk variants. This differs from the approach in Miguel-Escalada *et al* which prioritized genes of T2D risk loci in enhancers using chromatin conformation data yet did not further determine the effects of T2D risk variants in enhancers on gene activity. Therefore, while that study has greater resolution to identify candidate enhancer-promoter interactions, they do not consider the effects of enhancer

variants on these genes directly. We have clarified the approach we used in the abstract, introduction and main text:

Lines 43-45: “We fine-map T2D risk variants affecting islet enhancers, and find that candidate target genes of these variants defined using chromatin looping and eQTL mapping are enriched in protein transport and secretion pathways.”

Lines 72-75: “Candidate target genes of these T2D enhancer signals, defined by combining chromatin looping and promoter-proximity with eQTL mapping, are specifically enriched in protein secretion and transport pathways.”

Lines 268-272: “While a large percentage of T2D risk signals affect islet enhancer activity, the gene targets of these enhancers are unknown. In order to identify genes affected by T2D risk variants in enhancers, we used a tiered strategy whereby we first identified candidate target genes of these enhancers using chromatin looping and promoter-proximity, and then further prioritized candidate genes cis-regulated by T2D enhancer variants using eQTL mapping.”

27.) Lines 220-221: "Outside of known loci, we identified an additional 131 1Mb windows genome-wide harbouring putative T2D variants in islet enhancers."

How can the authors pinpoint variants within a 1 Mb window to specific enhancers?

We apologize for the confusion. We performed fine-mapping of T2D association data using variants within 1Mb windows genome-wide, through which we identified putative T2D risk variants at these loci. We then determined which of these putative T2D risk variants overlapped islet enhancers. We clarified this in the text by changing the description of these putative T2D variants:

Lines 238-242: “Outside of known loci, we identified an additional 127 loci genome-wide where fine-mapping identified a putative T2D risk variant that overlapped an islet enhancer and an ATAC-seq site from >1 sample (**Supplementary Figure 3C, Supplementary Data 1, Supplementary Table 7; see Methods).**”

28.) Figure 3H: abbreviations need to be explained in the figure legend (Alt = alternative; Ref = reference). Which one is the risk allele? According to the text, it would have to be the Ref allele, as this is the one driving increased luciferase activity? See also point 7 above.

We have clarified the legend for Figure 3H to define ‘alt’ and ‘ref’, and have also added to both the main text and the figure legend to clarify that the ‘ref’ allele is also the T2D risk allele for this variant.

Lines 258-260: “Among the 19 novel imbalanced variants, rs7732130 at 5p13 is causal for T2D (PPA=98%) and the T2D risk (and reference) allele G increased chromatin accessibility ($P=7.1 \times 10^{-4}$).”

Lines 1096-1098: “(H) rs7732130 has allelic effects on enhancer activity in the islet cell line MIN6 (N=3), where the T2D risk allele and reference (ref) G has higher activity than the alternate (alt) allele A.”

29.) Lines 185-187: "The effects of variants in regulatory elements in T2D risk in the context of chromatin looping, however, is unknown."

Should be: "The effects of variants in regulatory elements in T2D risk in the context of chromatin looping, however, are unknown."

We appreciate the reviewer's clarification and have updated the text accordingly.

Lines 200-202: "The effects of variants in regulatory elements on T2D risk in the context of chromatin looping, however, are unknown."

30.) Line 216-217: "...and at 6 signals resolved a single causal enhancer variant such as at ZBED3 (Figure 3G...)."

How was this enhancer variant linked with ZBED3? Using eQTL data? Chromatin looping data? Both? The enhancer seems to be located equidistant to ZBED3 and PDE8B.

We apologize for the confusion. It is commonplace to name GWAS risk loci by a nearby gene to facilitate cross-referencing the locus across studies, although of course in most cases the actual risk gene(s) are not known. In this case ZBED3 is simply the gene which has been previously used to reference this locus. We have now used more descriptive language in the main text in order to clarify this locus as 'the 5q13 locus near ZBED3/PDE8B', and at other places where this locus is referenced.

Lines 236-238: "At 6 signals we resolved a single causal enhancer variant, for example rs7732130 (PPA=98%) at the 5q13 locus near ZBED3/PDE8B (Figure 3G)."

We have also modified Figure 4B to reference the locus instead of a specific gene name for each T2D signal, as well as in Table 1, Supplementary Table 8 and Supplementary Figure 4.

31.) The genomic coordinates are missing for Figure 3G.

We have added the chromosome and genomic coordinates to Figure 3G.

32.) Lines 259-260: "...; for example, multiple KCNQ1 signals interacted with INS/IGF2 over 700 kb distal,..."

What exactly do the authors mean by 'KCNQ1 signals'? Sequence variants located in putative enhancers downstream of KCNQ1?

We again apologize for the confusion. There are multiple independent T2D risk signals at this locus, which all map proximal to KCNQ1. We have used more descriptive language in the text to clarify this as:

Line 283: "T2D enhancer variants at the 11p15 locus near KCNQ1"

As described in response to comment 30, we also updated references to this locus in Figure 4B, Supplementary Table 8 and Supplementary Figure 4 accordingly.

33.) Lines 260 - 261: "..., and ZMIZ1 interacted with POLR3A over 1MB distal." Again, the authors should use more precise language here. Are these sequence variants in introns of ZMIZ1?

We apologize for the confusion once again. We now refer to this T2D risk locus as:

Line 285: "T2D enhancer variants at the 10q22 locus near ZMIZ1"

As described in response to comment 30, we also updated references to this locus in Figure 4B, Supplementary Table 8 and Supplementary Figure 4 accordingly.

34.) Figure 4E: no chromosome number or sequence coordinates are provided.

We have added the chromosome and genomic coordinates to this figure panel (now Figure 4C in the revised manuscript).

35.) Lines 284-285; "..., and IGF2BP2 is the only implicated target gene at its respective locus in our analyses."

How has IGF2BP2 been implicated? On eQTL data and Hi-C data? Does the table in Figure 4D refer to Hi-C chromatin looping data (column '# target genes')? How far away is rs10428126 from the IGF2BP2 promoter? Is the Hi-C data resolution sufficient to link this sequence variant to the IGF2BP2 promoter?

We apologize for the confusion in the description of *IGF2BP2*. In table in Figure 4D (now Table 1 in the revised manuscript) the column '# candidate genes' refers to the number of genes where a T2D variant is either within 25kb of a chromatin loop to the gene promoter region or within 25kb of the gene promoter itself, which we describe in the main text:

Lines 268-274: "In order to identify genes affected by T2D risk variants in enhancers, we used a tiered strategy whereby we first identified candidate target genes of these enhancers using chromatin looping and promoter-proximity, and then further prioritized candidate genes cis-regulated by T2D enhancer variants using eQTL mapping. For each T2D enhancer signal (from Figure 3D), we identified candidate genes based on whether an enhancer variant was within 25kb of either a chromatin loop to the gene promoter or the gene promoter itself (see Methods)."

In the case of *IGF2BP2*, it is the only gene at its respective locus that satisfies either criterion, as rs10428126 is within 25kb of the gene promoter and not in a loop to any other gene promoter. There is also a strong eQTL for *IGF2BP2* co-localized with T2D variants at this locus (also shown in Table 1). Finally, *IGF2BP2* is the only gene promoter in the entire topologically-associating domain (TAD) containing the T2D variants at this locus. As this information was not included in the manuscript previously, we have added a new table that lists gene promoters contained within TAD boundaries genome-wide (Supplementary Data 4). Furthermore, we have also added a new Supplementary Figure 5A which shows the genomic context containing the TAD, *IGF2BP2* and the T2D risk variant. These results together strongly implicate *IGF2BP2* as a likely target gene of T2D risk variants at this locus. We have clarified the main text to reflect this:

Lines 324-327: “At the 3q27 locus IGF2BP2 is the only candidate target gene based on T2D variant proximity to the gene promoter and eQTL evidence (Table 1, Supplementary Figure 5A, Supplementary Table 8), and is furthermore the only gene promoter in the entire TAD (Supplementary Figure 5A, Supplementary Data 4).”

Lines 328-330: “Fine-mapped T2D enhancer variants at 3q27 all mapped within a 6kb intronic region proximal to the IGF2BP2 promoter (Supplementary Figure 5A, Supplementary Table 6).”

36.) Regrettably, the authors have not attempted to find a uniform font type or size for the figures

We have updated the figures to use a uniform font type and size.

Reviewer #2 (Remarks to the Author):

Greenwald et al have tried to respond to the majority of my comments. However, based on the small number of samples, which won't make their data representative to humans in general, I still do not find their manuscript suitable for Nature Communications. I believe their hi-C data needs to be replicated in independent samples, in a similar manner to for example GWAS. Also, I believe it would be good with expression data from their own cohort as well as additional cohorts.

Reviewer #4 (Remarks to the Author):

I was pleased to see that the authors have taken most of my suggestions on board (and provided clarification where I misunderstood), and have gone to great lengths to make extensive changes that have now resulted in a much-improved manuscript. I have a few minor suggestions (as detailed below), but once these are addressed I will be happy to recommend this manuscript for publication in Nature Communications.

Line 70: "..., and genes with enhancer interactions correlate..."

I think there is a verb missing here? "..., and find that genes with enhancer interactions correlate..."

Line 77: "...conditional knockout of IGF2BP2 (Imp2) in mouse islets..."

Clearer would be "...conditional knockout of IGF2BP2 homolog Imp2 in mouse islets..."

Line 128: "...were those in CTCF binding...states..."

Is there more than one CTCF binding CMM state?

Line 134: space after "OR="

Line 135: "...interactions between CTCF binding states..."

See above – there is only one CTCF CMM state. Would "...interactions between sites within the CTCF CMM state..." be more accurate?

Line 270: "...promoter-proximity..." Hyphen?

Line 298: “For example, known T2D variant...”

Is “For example, the known T2D variant...” better?

Lines 373/374: “In summary, we defined the genomic location, function, and spatial orientation of accessible chromatin in pancreatic islets.”

I think this should be toned down a little. I am ok with genomic location and spatial orientation, but the claim that the function has been defined for all these regions goes too far.

Line 379: “...promoter-proximity...” See above, hyphen?

Lines 384/385: “Furthermore, studies modifying islet enhancer activity through genome editing will provide additional validation...”

Maybe these experiments are already ongoing in the authors’ labs? If not, I would phrase this more carefully.

“Furthermore, studies modifying islet enhancer activity, for example through genome editing, may provide additional validation...”

Lines 389/390: “...that reduced activity of one of these target genes IGF2BP2 in mouse islets...”

This is not correct – it was reduced activity of the IGF2BP2 homolog *Imp2*.

Line 825: “ Mouse *Imp2*...” *Imp2* in italics

Line 1117: “Figure 5: Loss of IGF2BP2 activity...” Again, this should be *Imp2*. IGF2BP2 is not mentioned anywhere else in the figure legend or the figure itself, so keeping to *Imp2* throughout is not only correct, but will also make it easier for readers to understand this figure.

We thank the reviewers for their comments, and have addressed the comments from Reviewer #4.

REVIEWERS' COMMENTS:

Reviewer #2 (Remarks to the Author):

Greenwald et al have tried to respond to the majority of my comments. However, based on the small number of samples, which wont make their data representative to humans in general, I still do not find their manuscript suitable for Nature Communications. I believe their hi-C data needs to be replicated in independent samples, in a similar manner to for example GWAS. Also, I believe it would be good with expression data from their own cohort as well as additional cohorts.

Reviewer #4 (Remarks to the Author):

I was pleased to see that the authors have taken most of my suggestions on board (and provided clarification where I misunderstood), and have gone to great lengths to make extensive changes that have now resulted in a much-improved manuscript. I have a few minor suggestions (as detailed below), but once these are addressed I will be happy to recommend this manuscript for publication in Nature Communications.

We thank the reviewer for their comments, and have addressed all additional suggestions below

Line 70: "..., and genes with enhancer interactions correlate..."
I think there is a verb missing here? "..., and find that genes with enhancer interactions correlate..."

We have made the suggested change

Line 77: "...conditional knockout of IGF2BP2 (Imp2) in mouse islets..."
Clearer would be "...conditional knockout of IGF2BP2 homolog Imp2 in mouse islets..."

We have made the suggested change

Line 128: "...were those in CTCF binding...states..."
Is there more than one CTCF binding CMM state?

We have revised the text to clarify that there is one CTCF state

Line 134: space after "OR="

We have made the suggested change

Line 135: "...interactions between CTCF binding states..."

See above – there is only one CTCF CMM state. Would “...interactions between sites within the CTCF CMM state...” be more accurate?

We have clarified the text as suggested

Line 270: “...promoter-proximity...” Hyphen?

We have removed the hyphen

Line 298: “For example, known T2D variant...”
Is “For example, the known T2D variant...” better?

We have made the suggested change

Lines 373/374: “In summary, we defined the genomic location, function, and spatial orientation of accessible chromatin in pancreatic islets.”

I think this should be toned down a little. I am ok with genomic location and spatial orientation, but the claim that the function has been defined for all these regions goes too far.

We have modified the sentence accordingly:

“In summary, we defined the genomic location and spatial orientation of accessible chromatin in pancreatic islets.”

Line 379: “...promoter-proximity...” See above, hyphen?

We have removed the hyphen

Lines 384/385: “Furthermore, studies modifying islet enhancer activity through genome editing will provide additional validation...”

Maybe these experiments are already ongoing in the authors’ labs? If not, I would phrase this more carefully.

“Furthermore, studies modifying islet enhancer activity, for example through genome editing, may provide additional validation...”

We have made the suggested change

Lines 389/390: “...that reduced activity of one of these target genes IGF2BP2 in mouse islets...”

This is not correct – it was reduced activity of the IGF2BP2 homolog *Imp2*.

We have clarified that we observed the homolog *Imp2*, not IGF2BP2:

“Target genes of T2D islet enhancer signals were specifically enriched in protein transport and secretion pathways, and we validated that reduced activity of *IGF2BP2* homolog *Imp2* in mouse islets leads to defects in glucose-stimulated insulin secretion”

Line 825: “ Mouse *Imp2*...” *Imp2* in italics

We have made the suggested change

Line 1117: “Figure 5: Loss of IGF2BP2 activity...” Again, this should be *Imp2*. IGF2BP2 is not mentioned anywhere else in the figure legend or the figure itself, so keeping to *Imp2* throughout is not only correct, but will also make it easier for readers to understand this figure.

We have made the suggested change